# TarGEN: Targeted Data Generation with Large Language Models

**Himanshu Gupta**[1‡]    **Kevin Scaria**[1◇‡]    **Ujjwala Anantheswaran**[1◇∗]    **Shreyas Verma**[2]
**Mihir Parmar**[1]    **Saurabh Arjun Sawant**[1]    **Chitta Baral**[1]    **Swaroop Mishra**[1†]
[1]Arizona State University    [2]Georgia Institute of Technology
{hgupta35, kscaria}@asu.edu

## Abstract

We present TarGEN, a multi-step prompting strategy for generating high-quality synthetic datasets using LLMs. An advantage of TarGEN is its seedless nature; it does not require specific task instances, broadening its applicability beyond task replication. This differentiates it from other data generation techniques, as it can be leveraged for novel or highly domain-specific tasks with no existing data instances. We augment TarGEN with a *self-correction* module that enables LLMs to rectify inaccurately labeled instances during dataset creation, ensuring reliable labels. To assess our technique's effectiveness against existing baselines, we emulate eight tasks from the SuperGLUE benchmark to create a "synthetic" version and fine-tune various language models on both synthetic and original training sets. Evaluation on the original test set reveals that models trained on the synthetic data perform $\sim 1 - 3\%$ higher than those trained on original datasets Finally, when pre-finetuned on our "synthetic" SuperGLUE dataset, Llama2 (7B) yields impressive results on the OpenLLM leaderboard, surpassing the model trained on the Self-Instruct dataset by 2.62%. Our analysis reveals that the synthetic data generated by TarGEN not only improves model learning, but also has comparable or higher levels of complexity, diversity, and similar levels of bias in comparison with the original data [1].

## 1 Introduction

Large Language Models (LLMs) like ChatGPT, Llama (Touvron et al., 2023a;c), and Mistral (Jiang et al., 2023) have showcased impressive results across a plethora of tasks (OpenAI, 2023; Brown et al., 2020). As LLM capabilities advance, the tools to test the extent of these capabilities become insufficient (Liu et al., 2022b; He et al., 2023; Valmeekam et al., 2022; Chen et al., 2021). This is particularly true for domain-specific datasets, as the creation of expertly curated evaluation benchmarks is time and labor-intensive (Clark et al., 2018; Suzgun et al., 2022; Wang et al., 2022). Several synthetic dataset creation methods such as Self-Instruct (Wang et al., 2023), AttrPrompt (Yu et al., 2023) and ZeroGen (Ye et al., 2022a) have been proposed primarily for text classification tasks. These approaches employ in-context learning to generate synthetic data points that resemble their prompt exemplars, thereby inherently constraining their ability to produce diverse examples. Additionally, these approaches are often uniquely tailored to certain downstream tasks or datasets, reducing their adaptability. Consequently, they may be unsuited to synthesizing domain-specific tasks/benchmarks evaluating novel tasks where there is a lack of sufficient data samples to use as prompt exemplars.

To mitigate these issues, we introduce TarGEN, a multi-step prompting strategy (Fig 1). This approach consists of four steps. First, we initialize a set of contexts to inject semantic diversity, followed by the generation of task-specific elements we call "instance seeds" -

---

[1]Codebase: https://github.com/kevinscaria/TarGEN    ∗ Currently in Microsoft    † Currently in Google DeepMind    ‡ Currently in Amazon (The work was done prior to joining Amazon) ◇ Equal Contribution

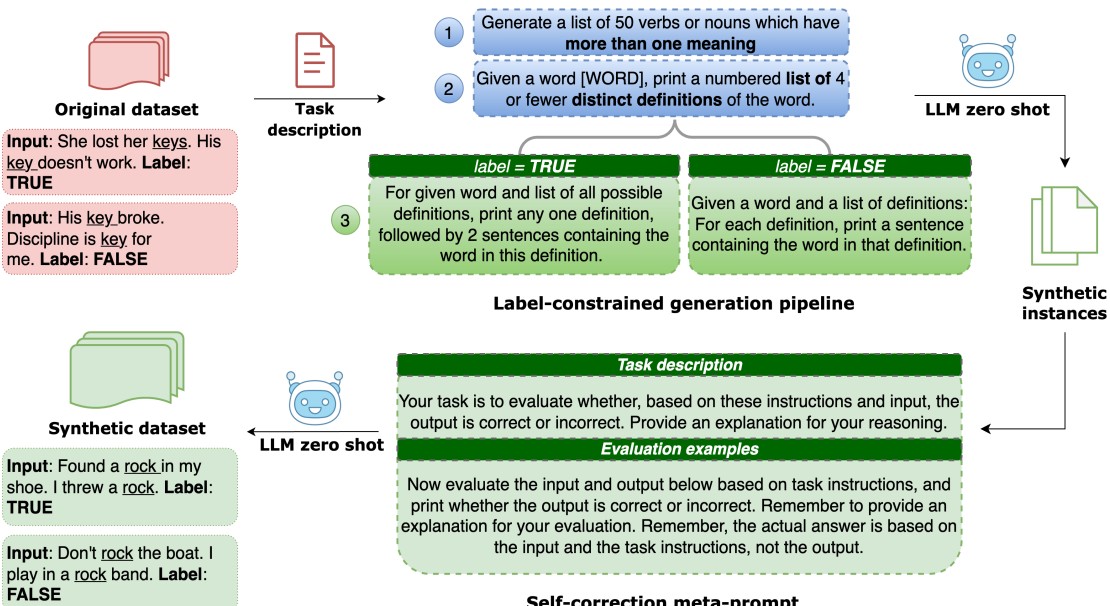

Figure 1: An overview of using TarGEN to generate instances for the WiC task. We first create a set of prompts (boxes 1, 2 in figure) to generate instance seeds, or linguistic components unique to each task instance. Next, we create label-specific prompts (box 3) that generate instances based on instance seeds and the relationship implied by the label for this task. We use zero-shot LLM inference to generate an initial set of synthetic instances. The instances are then passed to our *self-correction* module consisting of a single meta-prompt which allows us to re-label mislabeled data instances, helping us reduce noise. Hence, based on the task description, we obtain high-quality synthetic instances to evaluate a task.

elements that form the unique basis of each instance. These seeds can be sentences, passages, or more atomic elements but are not input exemplars. Then, for each "instance seed", we formulate a label constraint that uses these seeds to generate a data instance attributable to the constrained label.

Finally, we leverage our evaluator model for *self-correction* over the generated instances, and re-label them wherever necessary (§E). The self-correction module integrated into the pipeline contributes to noise reduction and enhances the overall quality of generated content. Using TarGEN, we can produce high-quality diverse datasets with precise labels. This method offers the benefit of not necessitating pre-existing task instances as starting points for generation. Consequently, it is suitable for domain-specific tasks and the generation of novel benchmarks, relying only on the provided task descriptions.

We use TarGEN in conjunction with ChatGPT to generate data for natural language tasks (textual entailment, question answering) used in benchmarks like SNLI, MNLI, and SciTail. To ensure fair evaluation against existing baselines, we pick tasks from the comprehensive and widely-recognized SuperGLUE benchmark and generate synthetic instances for them based solely on task descriptions. We treat each task as a standalone language understanding challenge, without directly referencing the benchmark or dataset name during generation. We train several models from different families (encoder only, encoder-decoder, and decoder only) on the synthetically generated train set and the original SuperGLUE train set and evaluate these models on the original test set. We find that models trained on the synthetic train set perform at par as opposed to models trained on the original train set. Instruction tuning results in a 3.42% increase for Flan T5 models and a 3.24% improvement for Pythia GPT models. We also observe that training on synthetic data in a multi-task fashion yields an average improvement of 4.73%, 3.21%, and 2.94% for T5-3B, Llama2-7B, and Mistral-7B respectively, over multi-task training on the original data (§4) The self-correction module plays a vital role in the improved performance; multi-task training with T5-3B on the self-

corrected synthetic data improves performance over non self-corrected synthetic data on all tasks by 5.9% on average. We also conduct a comparative study between TarGEN and Self-Instruct (Wang et al., 2023) by pre-finetuning Llama2 (7B) (Touvron et al., 2023c) on both datasets and evaluating them on the OpenLLM benchmark Our findings indicate that Llama2 pre-finetuned on our dataset outperforms Llama2 trained using Self-Instruct by 2.62% points.

A systematic human evaluation was conducted to appraise the quality of generated instances. Our thorough analysis of the data reveals the robustness of our synthetically generated SuperGLUE dataset in terms of dataset difficulty, diversity, and bias. It exhibits equivalent or higher dataset difficulty (lower $\mathcal{V}$-usable information (Ethayarajh et al., 2022)), showcasing the adequate complexity of our datasets. Furthermore, our datasets have comparable lexical diversity (Yu et al., 2023) and consistently display lower cosine similarity between intra-dataset text pairs, highlighting the datasets' rich and distinct content. Additionally, we measure the datasets' lexical diversity in terms of vocabulary count. Our results indicate that the datasets generated using TarGEN consistently exhibit higher lexical diversity than their original counterparts. In terms of bias, our datasets align closely with the original datasets, demonstrating a balanced representation of various named entities such as Geopolitical Entities (GPE), Person Entities, and Nationalities or Religious or Political groups (NORP). A detailed analysis is present in §5.

## 2 Related work

Recent research has witnessed the emergence of various methods harnessing LLMs as synthetic data generators (Long et al., 2024; Lu et al., 2023; Anantheswaran et al., 2024; Li et al., 2023a). Specifically, in the context of few-shot classification tasks where labeled data is scarce, several approaches have been introduced. SuperGEN (Schick & Schütze, 2021) and ZeroGEN (Meng et al., 2022a) leverage LLMs to produce synthetic data. For zero-shot tasks, SunGen (Gao et al., 2023) employs noise-filtering techniques, while ProGEN (Ye et al., 2022c) utilizes model feedback to ensure generated data quality. Similarly, (Chia et al., 2022) introduces structured prompts for tasks like relation triplet extraction. Moreover, (Liu et al., 2022a) and (Wiegreffe et al., 2022) propose synthetic data generation methods for natural language entailment (NLI) tasks and free-text explanations in a human-AI collaborative setting. There also have been approaches proposed to generate tabular data (Borisov et al., 2023) and instruction data (Peng et al., 2023; Sun et al., 2023).

The research direction in synthetic data generation predominantly focuses on zero/few-shot classification or entails finetuning (Chen et al., 2023) or iterative finetuning of open-source LLMs (Yu et al., 2023). In contrast, our method is simple, lightweight, and adaptable even to closed-source LLMs. Additionally, the aforementioned methods often rely on seed samples where expert annotators are employed to curate and label a sample set of considerable size. These seed samples are used to generate additional synthetic samples for various downstream tasks.

We exploit the generative capabilities of LLMs to completely generate seed instances that drive the diverse synthetic dataset generation. Furthermore, our approach uses a multi-step prompting strategy along with *self-correction* to for targeted data generation while preserving quality in terms of diversity, bias, noise, and mislabelling. Finally, existing dataset generation methods often are limited by reliance on seed tasks from the original dataset (Wang et al., 2023) whereas our seedless pipeline leverages high-level dataset characteristics for the generation process. We compare our framework with other popular data generation approaches along various axes in §A.1. For an empirical comparison with these approaches, see §A.2 of the Appendix.

## 3 TarGEN

### 3.1 Problem formulation

Given a dataset for a language task $t$, its data points can be expressed as $(d,l)$ such that there exists a function $f : \mathcal{D} \to \mathcal{L}$

$$f_t(d) = l, \forall d \in \mathcal{D}, l \in \mathcal{L} \tag{1}$$

where $f_t$ is a mathematical representation of the task $t$, $d \in \mathcal{D}$ is the instance input, and $l$ is the instance label from $\mathcal{L}$, the label space for the given task. We formalize dataset generation as a sequence of label-constrained text generation problems, where the generation of an instance is constrained by its label value. This allows us to control label distribution in our synthetic dataset and craft high-quality instances by clearly defining the relationships between instance and label.

This circumvents the need for any seed instances from the original dataset; i.e. for any given task, a dataset can be generated from scratch by formulating its generation function from the task description. The task-specific label-constrained dataset generation approach is given below:

| Datasets | Task Type | Instances Split |
|---|---|---|
| **AX-g** | NLI | Not Ent: 146 \| Ent: 138 |
| **BoolQ** | Bin. Class. | True: 2535 \| False: 1764 |
| **CB** | NLI | Cont: 119 \| Ent: 115 \| Neut: 16 |
| **COPA** | Bin. Class. | Choice 1: 195 \| Choice 2: 107 |
| **ReCoRD** | MCQ | 1778 MCQs |
| **RTE** | NLI | Not Ent: 1241 \| Ent: 1249 |
| **WiC** | Bin. Class. | True: 2433 \| False: 2410 |
| **WSC** | Bin. Class. | True: 259 \| False: 285 |

Table 1: Statistics of the dataset. Following abbreviations are used: Bin. Class: Binary Classification; Ent: Entailment, Cont: Contradiction, Neut: Neutral; NLI: Natural Language inference

$$\bigcup_{l \in \mathcal{L}} \bigcup_{n=1}^{N_l} (G_{l,t,n}(l, i_n), L = l) \tag{2}$$

where $N_l$ is the number of samples for the label $l$, $t$ is the task, and $G_{l,t}$ is an inverse function such that $f_t(G_{l,t}(l)) = l$, and $i_n$ is the $n^{th}$ instance seed. Formulating these task- and label-specific prompting strategies forms the crux of our simplified data synthesis pipeline. While these individual prompts are task-specific, the nature of these prompts, and the sequence they occur in, follow a framework engineered to create diversity and improve coverage. The stages of this framework are as follows:

**Step 1** We generate a set $\mathcal{C}$ of "contexts", or "settings" that provide unique semantic scope, such as "geopolitical news", "book review", or "movie script". These provide contexts within which a model can simulate a naturally occurring excerpt, to maintain semantic diversity.

**Step 2** We generate a set of passages, sentences, or task-specific elements, which we call "instance seeds". These seeds form the basis for each instance of the task.

**Step 3** We use the instance seeds as inputs to the generation prompt. This prompt is a descriptive formulation of the inverse generation function $G_{l,t}$ allowing us to generate data instances for the task.

**Step 4** We pass all generated data instances through a single-step *self-correction* process, where we use a task-specific prompt to reinforce the task instructions and identify and correct any mislabeled instances. This helps reduce noise and improve overall dataset quality.

### 3.2 Task-specific prompting strategies

The generation function $G_{l,t}$ for each task is included in §E. We generate a common set of contexts (Step 1) for almost all tasks.

| | AX-g | | | | BoolQ | | | |
|---|---|---|---|---|---|---|---|---|
| | Og | Syn | Og-I | Syn-I | Og | Syn | Og-I | Syn-I |
| **Cerebras** | 73.23 | 76.00 | 72.80 | 77.63 | 84.12 | 86.11 | 84.77 | 88.61 |
| **Pythia** | 74.90 | 78.43 | 74.57 | 79.80 | 83.29 | 84.02 | 82.61 | 85.05 |
| **T5** | 77.12 | 79.14 | 77.65 | 79.84 | 84.43 | 85.64 | 84.16 | 86.87 |
| **Flan** | 78.03 | 80.23 | 76.20 | 81.70 | 84.43 | 86.01 | 85.08 | 88.21 |
| **RoBERTa** | 77.10 | 78.17 | 76.35 | 82.14 | 84.43 | 84.56 | 84.07 | 87.56 |
| | COPA | | | | RTE | | | |
| **Cerebras** | 80.56 | 81.98 | 80.01 | 82.11 | 81.12 | 83.25 | 81.02 | 85.42 |
| **Pythia** | 79.98 | 80.74 | 80.54 | 83.27 | 82.98 | 83.65 | 82.85 | 87.87 |
| **T5** | 82.12 | 83.34 | 82.32 | 85.91 | 86.18 | 86.10 | 86.79 | 88.92 |
| **Flan** | 81.78 | 82.41 | 82.36 | 86.24 | 88.91 | 90.19 | 89.86 | 89.22 |
| **RoBERTa** | 82.12 | 83.98 | 80.99 | 83.16 | 88.20 | 89.01 | 88.32 | 88.84 |
| | CB | | | | ReCoRD | | | |
| **Cerebras** | 88.21 | 89.15 | 88.60 | 88.93 | 67.88 | 68.12 | 68.42 | 71.45 |
| **Pythia** | 91.93 | 89.63 | 89.74 | 93.03 | 68.83 | 69.19 | 68.61 | 73.36 |
| **T5** | 90.32 | 92.43 | 89.84 | 92.05 | 71.13 | 72.23 | 70.44 | 72.07 |
| **Flan** | 89.02 | 93.21 | 88.06 | 94.32 | 70.21 | 70.11 | 71.58 | 76.83 |
| **RoBERTa** | 87.86 | 90.12 | 92.36 | 90.48 | 69.88 | 70.21 | 70.29 | 70.02 |
| | WiC | | | | WSC | | | |
| **Cerebras** | 66.78 | 68.72 | 66.70 | 70.22 | 82.12 | 85.28 | 86.31 | 90.66 |
| **Pythia** | 68.33 | 71.77 | 70.53 | 70.84 | 86.71 | 87.32 | 86.44 | 88.09 |
| **T5** | 68.01 | 71.13 | 68.35 | 69.41 | 85.56 | 85.95 | 85.91 | 88.43 |
| **Flan** | 70.12 | 72.45 | 69.29 | 71.43 | 86.23 | 88.13 | 88.08 | 88.35 |
| **RoBERTa** | 69.90 | 70.06 | 70.16 | 71.23 | 87.21 | 85.65 | 82.37 | 83.21 |

Table 2: Performance of various models over SuperGLUE tasks. For each dataset, we compare the performance of these models (Cerebras, Pythia, T5, FLAN, RoBERTa) when trained on 4 distinct variants: **Og** (original train data), **Syn** (synthetic train data), **Og-I** (Instruction tuning on original data), **Syn-I** (Instruction tuning on synthetic data). For each task, we denote the highest performing model trained on original train data in green, and the highest performing model trained on synthetic train data in blue. All results are presented in %. We measure performance in accuracy, except in the case of ReCoRD, where we use the Rouge-L score.

**Dataset Statistics:** We choose the following tasks from the SuperGLUE benchmark: 1. CommitmentBank (CB). 2. Choice of Plausible Alternatives (COPA). 3. Recognizing Textual Entailment (RTE). 4. Word-in-Context (WiC). 5. Winograd Schema Challenge (WSC). 6. BoolQ. 7. Reading Comprehension with Commonsense Reasoning (ReCoRD). and 8. Wino-gender Diagnostics (AX-g). Given the challenges posed by each dataset, we employ TarGEN to create synthetic instances that can be used to evaluate language model performance. Table 1 showcases the instances split used in the original and synthetic datasets. Since label generation is controlled, synthetic SuperGLUE was created to match the exact number of original instances, while maintaining a balanced label distribution[2]. The exact data synthesis pipeline and prompts used for each dataset can be found in §E.

**CB** (De Marneffe et al., 2019): We generate sample pairs that share the relationship between label $l \in \{entailment, neutral, contradiction\}$, and based on a given context $c \in \mathcal{C}$.

**COPA** (Roemmele et al., 2011): We generate instances for CAUSE and EFFECT relations. In this case, Step 3 involves generating a premise and 2 hypotheses for a given context. For each relationship $r$, we generate (1) sentence pairs $(P, C)$ such that the premise and hypothesis

---

[2]Due to ChatGPT budget constraint, ReCoRD dataset was truncated to have 1778 instances, BoolQ dataset had 4299 instances and MultiRC dataset was skipped.

| Model | BoolQ | WiC | CB | AX-g | ReCoRD | RTE | WSC | COPA |
|---|---|---|---|---|---|---|---|---|
| **T5-3B Og** | 89.02 | 68.42 | 84.54 | 41.66 | 68.52 | 87.72 | 72.11 | 94.65 |
| **T5-3B Syn** | 90.04 | 74.78 | 93.22 | 49.13 | 76.59 | 92.41 | 73.07 | 95.20 |
| **Llama2-7B Og** | 88.51 | 72.26 | 87.66 | 48.71 | 72.34 | 88.76 | 75.43 | 94.85 |
| **Llama2-7B Syn** | 90.12 | 73.21 | 94.87 | 51.32 | 78.81 | 92.89 | 77.32 | 95.72 |
| **Mistral-7B Og** | 89.91 | 74.32 | 88.92 | 50.34 | 76.87 | 89.97 | 74.34 | 91.92 |
| **Mistral-7B Syn** | 90.81 | 75.95 | 94.21 | 54.19 | 80.17 | 93.09 | 78.21 | 93.43 |

Table 3: Results on T5-3B, Llama2-7B, and Mistral-7B, trained in a multi-task fashion. Og: Using a combined set of original datasets to train the model. Syn: Using synthetic versions to train the model. All numbers are in %. For each task, we denote the highest performing model trained on original train data in green, and the highest performing model trained on synthetic train data in blue.

| | BoolQ | WiC | CB | AX-g | ReCoRD | RTE | WSC | COPA |
|---|---|---|---|---|---|---|---|---|
| **T5-3B w/o SC** | 85.32 | 69.94 | 91.09 | 35.28 | 72.38 | 87.65 | 63.26 | 92.45 |
| **T5-3B with SC** | **90.04** | **74.78** | **93.22** | **49.13** | **76.59** | **92.41** | **73.07** | **95.26** |

Table 4: Results using a T5-3B model without and with the self-correction (SC) module. We observe that using self-correction improves performance over all datasets by 5.9% on average.

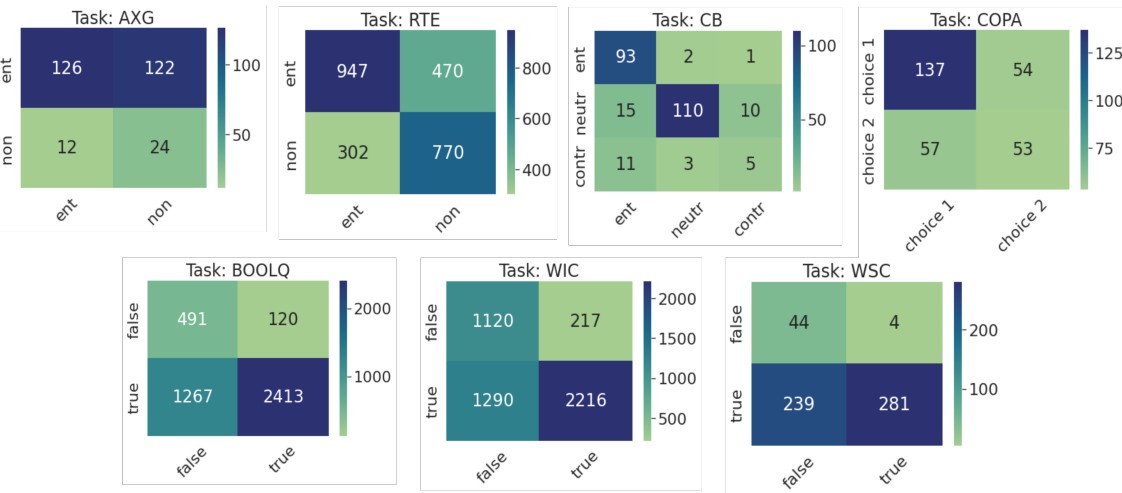

Figure 2: Matrices showing the effect of the *self-correction* step across various datasets of SuperGLUE. The row values show the number of labels that were originally assigned to that label (ent, non: entailment, non-entailment; neutr, contr: neutral, contradiction). The number in a cell $(i, j)$ reflects the number of labels originally assigned to label $i$ which were re-labeled to label $j$ after self-correction. While the majority of the instances had their labels reaffirmed by self-correction, a significant number of instances were re-labeled as a result of this step.

share the relationship specified, and (2) an alternate hypothesis $C_{alt}$ which explicitly does not share the specified relationship with the premise $P$. Thus $(P, C) \in r, (P, C_{alt}) \notin r$. The label space for this task is defined as $\mathcal{L} = \{C1, C2\}$. To ensure an even label split, we alternatively select attribute instances of $C$ as $C1$ and $C_{alt}$ as $C2$ and vice versa.

**RTE**: A collection of textual entailment challenges. For this task, step 2 consists of generating a set of premises $\mathcal{P}$ as instance seeds. For each $p \in \mathcal{P}$, we then generate hypotheses that are either logically sound ($l = entailment$) or logically unsound, i.e. do not follow the premise ($l = notentailment$).

**WiC** (Pilehvar & Camacho-Collados, 2019) : We curate a list of homonyms ($\mathcal{S}$) and all their definitions ($\mathcal{M}_s \forall s \in \mathcal{S}$). These act as instance seeds. For each $m \in \mathcal{M}_s$, given the label *True*, we generate a pair of sentences $(d1, d2)$ containing the word $s$, such that the definition of $s$ in $d1$ and $d2$ is $m$. For the label *False*, we randomly choose $m1, m2 \in M_s, m1 \neq m2$ and generate $d1, d2$ such that the definitions of $s$ in $d1$ and $d2$ are $m1$ and $m2$ respectively, making them distinct in word sense.

**WSC** (Levesque et al., 2012): For each context, we generate pairs of distinct noun phrases $(N_1, N_2)$ with identical plurality. This pronoun and noun phrases act as instance seeds. For each pair $(N1, N2)$, we then generate text $s$ containing $N1$ and $N2$ with all pronouns labeled with coreferred noun phrases. We randomly select an ambiguously-coreferent pronoun $P$ from the text. Based on this pronoun and the label constraint, we affix a noun phrase to the input instance.

**BoolQ** (Clark et al., 2019): For this task, we generate passages with multiple entities and inter-entity relations which act as instance seeds in Step 2. In Step 3, we generate a query, based on the passage $p \in \mathcal{P}$ and the label constraint $l \in \{Yes, No\}$. In the case of $l = Yes$, the query is information that can be inferred from the passage. For $l = No$, we generate a query that contradicts the passage.

**ReCoRD** (Zhang et al., 2018): This dataset evaluates understanding implied entailment relationships. In Step 2, we generate a set of passages $\mathcal{A}$ to act as instance seeds. Next, for each article $a \in \mathcal{A}$, we generate a complex, context-relevant sentence, and subsequently obscure a single entity reference.

**AX-g** (Rudinger et al., 2018): For this task, we generate 10 subject pairs that are from the same domain space, in Step 2. For each subject pair, we then generate an independent clause containing these subjects. These independent clauses are then used to generate dependent clauses coreferent with each subject, to act as instance seeds. In step 3, we use these subject-specific dependent clauses and the subject pairs to generate gender-agnostic hypotheses based on label constraints.

|         | Og    | Syn   | Og-I  | Syn-I |
|---------|-------|-------|-------|-------|
| **Cerebras** | 78.64 | **79.83** | 78.80 | **81.88** |
| **Pythia**   | 79.48 | **80.59** | 79.49 | **82.66** |
| **T5**       | 80.72 | **82.00** | 80.83 | **82.94** |
| **Flan**     | 81.12 | **82.84** | 81.49 | **84.54** |
| **RoBERTa**  | 79.98 | **81.47** | 80.08 | **82.08** |

Table 5: Avg. single-task performance across tasks, models, and data variants. Og, Syn, and I represent Original, Synthetic, and Instruction tuning respectively.

|             | L2SSG | L2SI  |
|-------------|-------|-------|
| **ARC**       | 54.65 | 52.30 |
| **HellaSwag**  | 79.25 | 77.09 |
| **MMLU**       | 45.63 | 41.60 |
| **TruthFulQA** | 42.34 | 37.58 |
| **Winogrande** | 71.37 | 69.46 |
| **GSM8K**      | 1.98  | 1.44  |
| **Average**    | **49.20** | **46.58** |

Table 6: Comparison of Synthetic SuperGLUE, Self Instruct Dataset, and AttrPrompt. T5-3B pre-finetuned on both datasets individually and finetuned on OpenLLM datasets.

### 3.3 Self-correction

Despite learning capabilities, LLMs demonstrate inconsistent reasoning (Ye & Durrett, 2022). We remedy them by implementing *self-correction*, an evaluation strategy that corrects inconsistent labels in the data synthesis process. We leverage an LLM as an evaluator model (ChatGPT in this case) to verify the alignment between the generated instances and their labels, as well as the alignment between these instances and the task description. *Self-correction* consists of a single meta-prompt that is common to all tasks. The task instructions and task-specific validation examples are used to augment the meta-prompt and tailor it to each generated dataset. This meta-prompt and the task instructions are in §G. Based on the provided input, the meta-prompt helps evaluate the correctness of its attributed label. If this label is deemed not correct, the evaluator model generates the correct label based on the instructions and instance input. Notably, from Fig 2 it can be seen that a significant number of instances require relabeling especially for high complexity tasks such as AX-g and WSC.

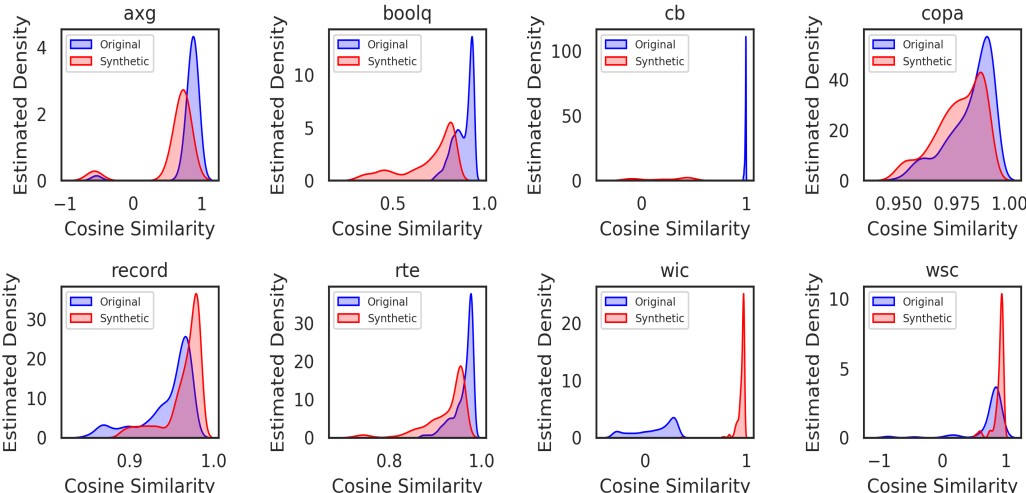

Figure 3: Comparison of semantic diversity across datasets among the original and the synthetically generated dataset. It can be seen that the original datasets' cosine similarity is higher for most tasks as compared to the synthetic datasets' which has a consistently lower cosine similarity indicating higher semantic diversity.

| Dataset | BoolQ | WiC | CB | AX-g | ReCoRD | RTE | WSC | COPA |
|---|---|---|---|---|---|---|---|---|
| Original | 251.8k | 65.8k | 6.4k | 3.2k | 160.3k | 79.1k | 8.8k | 5.1k |
| Synthetic | 190.6k | 76.4k | 16.6k | 4.4k | 236.7k | 49.4k | 8.9k | 5.2k |

Table 7: Lexical diversity of the dataset. Figures in red correspond where diversity of synthetic dataset is limited. Blue correspond where the synthetic dataset has more diversity than original.

## 4 Experiments and results

We train five models in the single-task learning (STL) setting for each original and synthetic dataset separately. We evaluate them on the original test set. The above experiments are repeated with instruction tuning as well. We also perform a multi-task learning (MTL) experiment where three models are finetuned in a multi-task fashion on the original and synthetic datasets separately (Mishra et al., 2021). The results in §4.1 are the average of five runs. **Models:** We use RoBERTa Large (354M) (Liu et al., 2019), Pythia GPT (410M) (Biderman et al., 2023), Cerebras GPT (590M) (Dey et al., 2023), Flan T5 Large (780 M) (Chung et al., 2022), T5 Large (780M) (Raffel et al., 2020) in STL setting, and T5-3B, Llama2-7B (Touvron et al., 2023b), and Mistral-7B (Jiang et al., 2023) in the MTL setting.

**Evaluation metric:** Following the SuperGLUE benchmark, we use accuracy to measure performance over all tasks, except for ReCoRD, which we evaluate on Rouge-L score.

### 4.1 Results

Table 2 showcases the results of models trained on both original and synthetic datasets for each task. Models finetuned on synthetic data consistently match or outperform variants finetuned on the original data. Furthermore, performance improves by 3% by instruction tuning (Mishra et al., 2021; Wei et al., 2021; Scaria et al., 2023; Gupta et al., 2023).

Table 5 gives the average model-wise results for the same. Table 3 denotes the results when the datasets are trained in multitask fashion with larger models. Table 4 highlights the effect of self correction module when T5 3B is trained without self correction module.

## 5 Analysis

In this section, we present a comprehensive analysis of the synthetic data generated, examining both quantitative and qualitative aspects.

**Human Evaluation**  As part of the quality evaluation, a systematic human evaluation was carried out over all the generated datasets to the best of our abilities. To ensure full coverage we performed human evaluation over 20% of all synthetic datasets. The evaluation was conducted equitably by 3 teams of 2 evaluators each. These teams comprised the authors, and thus required no external labour. The generated instances were graded on a scale of A (excellent) to D (poor). The evaluators graded the generated instances within the context of the original label constraint, as well as within the context of the updated labels after self-correction. We notice that as the instances go through self-correction, the number of samples rated "excellent" increases and the number of "unacceptable" samples reduces. To gauge the reliability of our human evaluation approach, we measure inter-evaluator agreement using the Cohen's Kappa (resulting in K = 0.71, 0.67, 0.79) and Spearman's correlation coefficient (obtaining r = 0.85, 0.83, 0.91). Details can be found in §I. We also found that the synthetic data contained no instances from the original SuperGLUE dataset, preventing any data leakage.

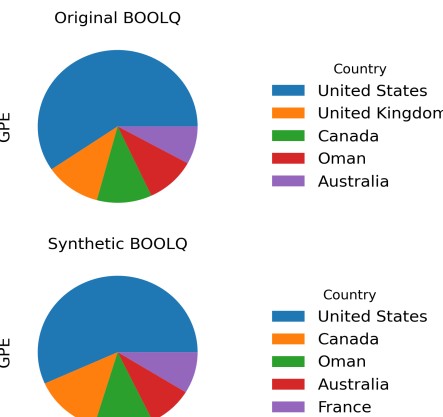

Figure 4: Comparison of dataset bias for the BoolQ dataset and the synthetically generated BoolQ dataset. This represents the distribution for GPE (Geo Political Entitiy) named entity tag. The distribution for other entities can be found in §C

**Comparison with Self-Instruct:** To compare the synthetic SuperGLUE dataset with other synthetic instruction following benchmarks, we choose Self-Instruct (Wang et al., 2023) a popular synthetic dataset generation framework. We choose Llama2 (7B) and pre-finetune using synthetic SuperGLUE. We refer to this tuned model as L2SSG. We refer to the Llama2 model trained on the Self-Instruct dataset as L2SI. Both models are evaluated on OpenLLM benchmark. Table 6 shows that L2SSG scores 2.62% higher than L2SI, underscoring the quality of targeted dataset generation by the TarGEN framework over Self-Instruct.

**Lexical diversity:** We analyze the dataset diversity, along the lines presented in (Yu et al., 2023). We initiate our exploration with a straightforward vocabulary-based examination to assess lexical diversity, as summarized in Table 7. Notably, our TarGEN synthetic data exhibits an average lexical diversity that is 25% higher across various dataset tasks.

**Semantic diversity:** To analyze the semantic diversity, we examine the cosine similarity distribution of SentenceBERT embeddings of within-dataset sample pairs as presented in Fig 3. Across most SuperGLUE tasks, the TarGEN datasets consistently display lower cosine similarity than the original dataset, indicating reduced semantic similarity of within-dataset samples and, consequently, higher semantic diversity. This observation underscores our approach's intrinsic capability to generate diverse samples. Moreover, our findings of higher cosine similarity of the original datasets align with those of (Parmar et al., 2023), where the authors highlight the propensity for crowdsourced datasets to exhibit high bias and low diversity. This phenomenon arises as crowdsourced workers often adhere to patterns provided by dataset creators. The dataset generated using our approach can generate high-diversity samples.

**Dataset difficulty:** To estimate the dataset difficulty, we use $\mathcal{V}$-usable information (Ethayarajh et al., 2022), which estimates the information an input $X$ holds for predicting the target $Y$ across a family of models $\mathcal{V}$. Lower $\mathcal{V}$-usable information indicates higher dataset difficulty for $\mathcal{V}$. Fig. 5 offers a comparative view of dataset difficulty between the original and synthetically generated datasets by TarGEN. Notably, the synthetic datasets exhibit a diverse range of samples with varying pointwise $\mathcal{V}$-usable information, showcasing their

diversity in terms of difficulty. Furthermore, the absence of mislabeled samples, indicated by positive $\mathcal{V}$-usable information in the synthetically generated datasets, underscores the effectiveness of *self-correction* prompts.

**Dataset bias:** To estiamte the dataset bias, we visualize the distribution of tokens related to named entities, belonging to Geopolitical Entities (GPE), Products, and Nationalities or Religious or Political groups (NORP). The distribution of input tokens for the original and synthetically generated BoolQ dataset is presented in Fig. 4. Our results reveal that the distribution of GPE, Product, and NORP entities in the TarGEN dataset closely resembles that of the original dataset. [3]. Additional experiments and extended analysis (including dataset comparison plots) of our approach can be found in §B and §C of the Appendix.

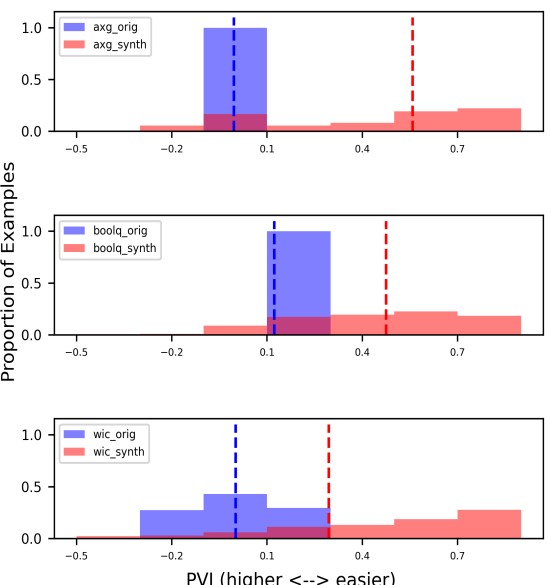

Figure 5: Comparison of PVI ($\mathcal{V}$-usable information) for AXG, BoolQ, and WiC original and the synthetically generated dataset. Synthetic data seems to have better quality as the original datasets' PVI is concentrated around -0.1 to 0.1 whereas the synthetic data generated has a diverse mix of difficulty level among the samples. The dataset difficulty comparison for all datasets can be found in §6

## 6 Conclusion

In this work, we introduced TarGEN, a multi-step prompting strategy for generating high-quality and diverse synthetic datasets utilizing LLMs without any human supervision.

We described a step-by-step methodology for TarGEN to synthesize a dataset from instructions without any task exemplars. To evaluate our proposed framework, we emulated eight tasks from the SuperGLUE benchmark and compared it with the original SuperGLUE by training different families of models. Experimental results reveal that models finetuned on our synthetic SuperGLUE outperform models finetuned on the original SuperGLUE. A comprehensive analysis of synthetic benchmark with respect to the original benchmark resulted in several interesting findings such as the fact that data instances in our synthesized benchmark are more difficult and diverse compared to the original benchmark, and also exhibit similar dataset bias.

## Limitations

Comparison with Self-Instruct and AttrPrompt revealed that synthetic SuperGLUE served as better pre-finetuning corpora when evaluated on the OpenLLM benchmark resulting in an impressive performance when using T5-3B and Llama2-7B. Though TarGEN facilitates high-quality data generation, we believe that it is important to assess our proposed frameworks within a multi-lingual context and also on additional benchmarks, including BigBench, LILA, and HELM. Furthermore, TarGEN currently relies on the ChatGPT model for synthesizing the benchmark, but our plans involve exploring the impact of other LLMs such as GPT-4, Llama2, and Falcon when employed with TarGEN. We believe that TarGEN can serve as a valuable tool for enhancing the quality of data generation, thus reducing human effort.

---

[3]To ensure a broad-scale examination across all samples, we utilized the Spacy library and its *en_core_web_sm* model to extract named-entity tags.

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

# Appendix

# A Comparison with other data generation frameworks

## A.1 Qualitative comparison

| Method | Seedless | Training required as part of pipeline | Focused diversity | Ability to generate samples for a complex task | Noise mitigation strategy |
|---|---|---|---|---|---|
| Self Instruct Wang et al. (2023) | No | No | No | No | Yes |
| CODA Evuru et al. (2024) | No | No | No | No | No |
| Synthetic Data Generation Li et al. (2023b) | Yes | No | No | No | No |
| ZeroGEN Ye et al. (2022b) | Yes | Yes | No | No | No |
| SuperGEN Meng et al. (2022b) | Yes | Yes | No | No | No |
| SunGEN Gao et al. | Yes | Yes | No | No | Yes |
| ProGEN Ye et al. (2022d) | Yes | Yes | No | No | Yes |
| Ours (TarGEN) | Yes | No | Yes | Yes | Yes |

Table 8: Qualitative comparison with previous works along different aspects of methodology.

We compare our framework with other popular data generation approaches in the table 8. Axes of comparison:

- **Seedless:** Whether the approach requires labeled task samples.
- **Focused diversity:** Whether the approach actively injects diversity, incentivizing samples generated across various contexts.
- **Noise mitigation strategy:** Whether the approaches actively mitigate noise, either algorithmically or through the inclusion of a module like our Self-Correction as part of their pipeline.

## A.2 Empirical comparison

To show a quantitative comparison with other data generation methods, we generate synthetic versions of specific tasks from the SuperGLUE dataset using recent data generation frameworks: The tasks we generate are RTE (textual entailment) and AXg (gender disambiguation). We then train a Llama-2 model on these synthetic task variants and evaluate them on original test sets of the datasets. The results are present in the Table 9.

| Method | RTE | AXg |
|---|---|---|
| CODA | 64.56 | 23.46 |
| ZeroGEN | 58.54 | 19.01 |
| SuperGEN | 77.94 | 44.67 |
| SunGEN | 75.11 | 38.16 |
| ProGEN | 78.43 | 37.94 |
| Ours | **92.89** | **51.32** |

Table 9: Results on RTE, Axg for Llama-2 models trained on synthetic data generated by existing data generation frameworks.

# B Additional Experiments

## B.1 Prompt variations on different downstream tasks

To understand the effect of prompt variations on different downstream tasks, we generated 2 additional variations of the Step 3 prompt for CommitmentBank dataset. Llama-2 (7B)

was fine-tuned on these alternative synthetic datasets alongside our initially-generated synthetic dataset and evaluated on the original test set. We use the following prompts for this experiment:

**Variation 1**

- *The input contains a premise and a hypothesis.*
- *If the hypothesis logically follows the premise and the hypothesis can be derived from the information in the premise, Output "entailment"*
- *If the hypothesis directly contradicts information in the premise, Output "contradiction"*
- *If the hypothesis is unrelated to the premise, or cannot be sufficiently proven from the information in the premise, Output "neutral"*

**Variation 2**

- *The input consists of a premise and a hypothesis.*
- *Output "neutral" if the hypothesis is either unrelated to the premise or cannot be conclusively derived from it.*
- *Output "contradiction" if the hypothesis directly opposes the information in the premise.*
- *Output "entailment" if the hypothesis logically follows from and can be deduced from the premise.*

**Variation 3**

- *The input consists of a premise and a hypothesis.*
- *If the hypothesis is directly opposed to the information in the premise, output "contradiction."*
- *If the hypothesis logically follows from the premise and can be inferred from its information, output "entailment."*
- *If the hypothesis is unrelated to the premise or lacks sufficient evidence from the premise, output "neutral."*

| Dataset | Mean Cosine Similarity | Accuracy Llama-2 (7B) |
|---|---|---|
| **Variation 1** | 0.56 | 94.85 |
| **Variation 2** | 0.42 | 93.27 |
| **Variation 3** | 0.47 | 94.14 |

Table 10: Accuracy of inference results for the original Step 3 prompt and 2 variations. Mean cosine similarity: Average of cosine similarity between given prompt and the other variations.

We see from the results in Table 10 that there is slight variation in the final results and similar mean cosine similarity. We would also like to highlight that the motivation behind proposing a clearly-defined, task-agnostic guiding framework such as TarGEN is to constrain the prompt generation to a series of simpler, clearly defined subtasks. The cascaded stepwise approach has the following advantage: At each step, the subtask for which we prompt the model is straightforward with very little room for misinterpretation and errors. This reduces the prompt engineering effort significantly, especially compared to the human cost of engineering a single prompt that completely and accurately models a complex task. Any prompt that adheres to the instructions for that particular step in the framework is likely to generate similar outcomes, making this framework task- and prompt- agnostic.

Additionally - the constrained, straightforward nature of the task prompt at each step makes it less likely that any variance in prompt would cause a significant change in outcome,

given the narrower scope of the objective at each step and the fact that the outcomes are grounded in the response to the previous step. Furthermore, any slight deviances could be attributable to the randomness inherent in inferring with LLMs, but we would not expect to see a significant difference in quality or diversity of the output samples.

## B.2 Effect of seed instances

For the seed instances, we experimented to see the impact of reducing the number of semantic contexts. Table 11 shows the effect of the generating samples over a proportion of the original semantic contexts:

| Scores with | BoolQ | WiC | CB | AX-g | ReCoRD | RTE | WSC | COPA |
|---|---|---|---|---|---|---|---|---|
| **100%** | 90.12 | 73.21 | 94.87 | 51.32 | 78.81 | 92.89 | 77.32 | 95.72 |
| **50%** | 85.43 | 72.65 | 90.33 | 49.86 | 75.42 | 89.75 | 76.21 | 92.11 |
| **25%** | 83.65 | 71.78 | 88.47 | 48.32 | 73.78 | 87.81 | 76.02 | 91.21 |

Table 11: Accuracy of models trained over samples generated over the full (100%), half, and a quarter of the original semantic contexts

## B.3 Comparison with other generative models

We run the data generation pipeline with Claude Sonnet and Llama-3 (70B) to create 2 more variants of synthetic SuperGLUE with each model keeping the task set, prompts, and pipeline steps constant. Llama-2 (7B) was fine-tuned on the above-mentioned datasets and evaluated on the original test set. The results of the experiments are attached in Table 12. We see nearly the same or slightly improved scores when using Llama-3 (70B) (Dubey et al., 2024) and Claude Sonnet (The).

| Train data Models | BoolQ | WiC | CB | AX-g | ReCoRD | RTE | WSC | COPA |
|---|---|---|---|---|---|---|---|---|
| **ChatGPT (current)** | 90.12 | 73.21 | 94.87 | 51.32 | 78.81 | 92.89 | 77.32 | 95.72 |
| **Claude Sonnet** | 90.65 | 73.88 | 95.23 | 51.88 | 79.22 | 93.01 | 77.89 | 95.72 |
| **Llama-3 (70B)** | 89.56 | 73.33 | 93.65 | 51.88 | 79.01 | 92.23 | 77.65 | 95.23 |

Table 12: We observe that the data generation pipeline is adaptable to different generative models with negligible performance degradation.

# C Extended Analysis

## C.1 Dataset difficulty comparison

The section contains plots pertaining dataset difficulty and bias for all datasets.

## C.2 Dataset bias comparison

## C.3 Generalizability of the framework

The basic framework proposed in this work is task and model-agnostic. While RTE is a relatively straightforward task, AXg is a complex NLU task. Despite the differences in the prompts of these tasks, the underlying TarGEN framework delineating the steps remains common. This proposed framework can be adapted to any task, based on the problem being evaluated. Given the parameters of a task - the label schema, and the problem statement, this framework is easy to adapt for multiple tasks of varying complexities. We would like to highlight that the simple nature of our framework allows for greater flexibility and control, making it well-suited to domain-specific or particularly complex tasks. We are happy to generate any other tasks that the reviewer deems necessary for the verification of this approach.

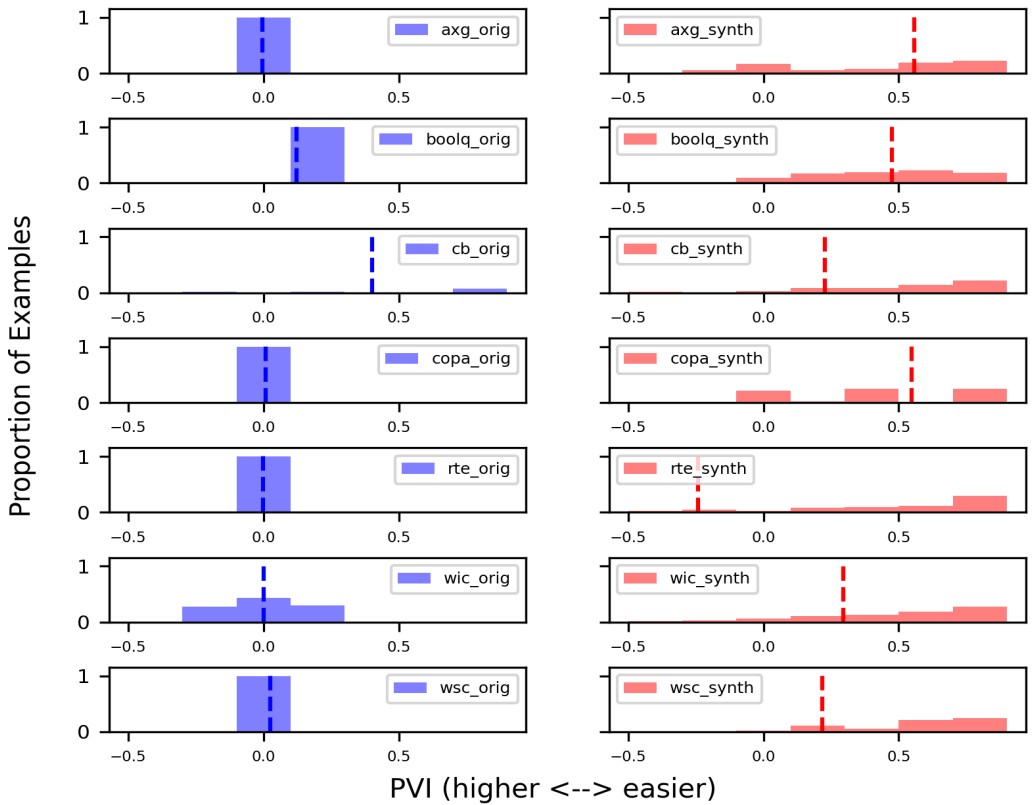

Figure 6: Comparison of PVI ($\mathcal{V}$-usable information) across datasets for the original dataset and the synthetically generated dataset. Synthetic data seems to have better quality as Original datasets PVI is concentrated around -0.1 to 0.1 whereas the synthetic data generated has a diverse mix of difficulty level among the samples.

### C.4 Regarding $\mathcal{V}$-Usable Information

$H_r(Y) - H_r(Y|X)$, where $H_r(Y)$ is the predictive entropy from a null model $LM_{Null}$ (trained on empty inputs), and $H_r(Y|X)$ is the conditional entropy from a model $LM_X$ (trained on input samples and labels). Entropies are measured in bits using $\log_2(P)$, where $P$ is the probability distribution of the labels assigned by the specific model. Higher $\mathcal{V}$-usable information indicates harder samples, while lower values indicate easier samples for model family $\mathcal{V}$. Original datasets have $\mathcal{V}$-usable information concentrated in a narrow range of easier samples. Our strategy generates a broader range of $\mathcal{V}$-usable information, from easy to hard, leading to better finetuning results.

## D Hyperparameters

We use 6xNvidia Tesla P40 GPUs. Batch Size: 16 for STL and 1 MTL setting. Gradient Accumulation Steps: 1, Learning rate: 5e-5, Num of Epochs: 5 for STL and 1 MTL setting.

### D.1 Additional Results

## E Task-specific instance generation pipelines

This section details the task-specific TarGEN strategy used to generate instances.

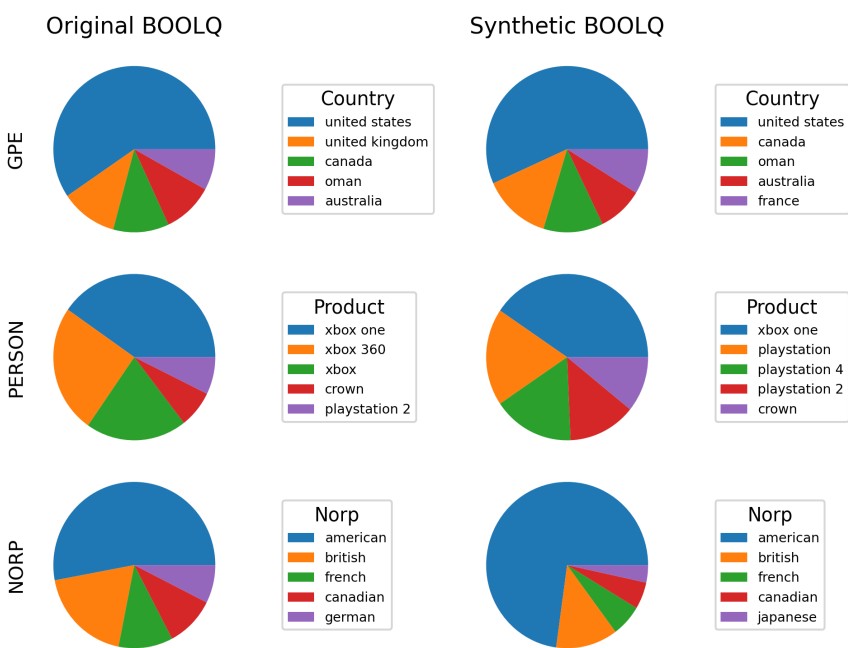

Figure 7: Comparison of dataset bias for the BoolQ dataset and the synthetically generated BoolQ dataset.

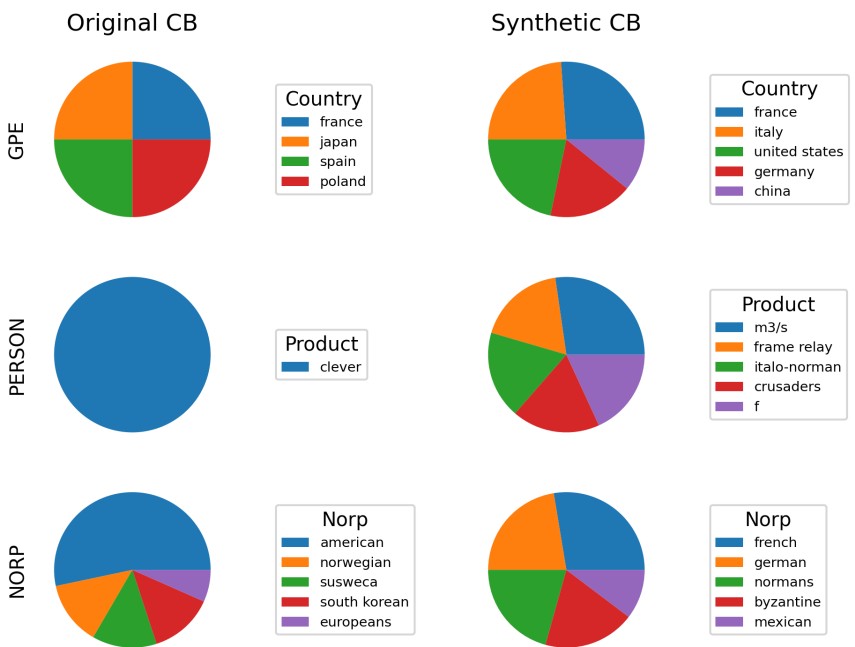

Figure 8: Comparison of dataset bias for the CB dataset and the synthetically generated CB dataset.

## E.1 COPA

*Step 1: Generate a list of domains or settings in which events can take place.*

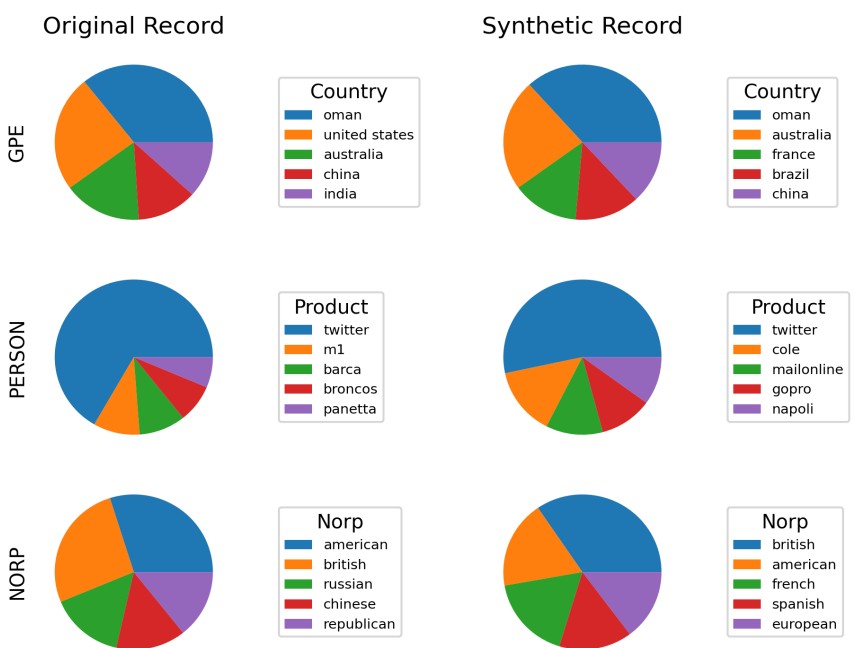

Figure 9: Comparison of dataset bias for the Record dataset and the synthetically generated Record dataset.

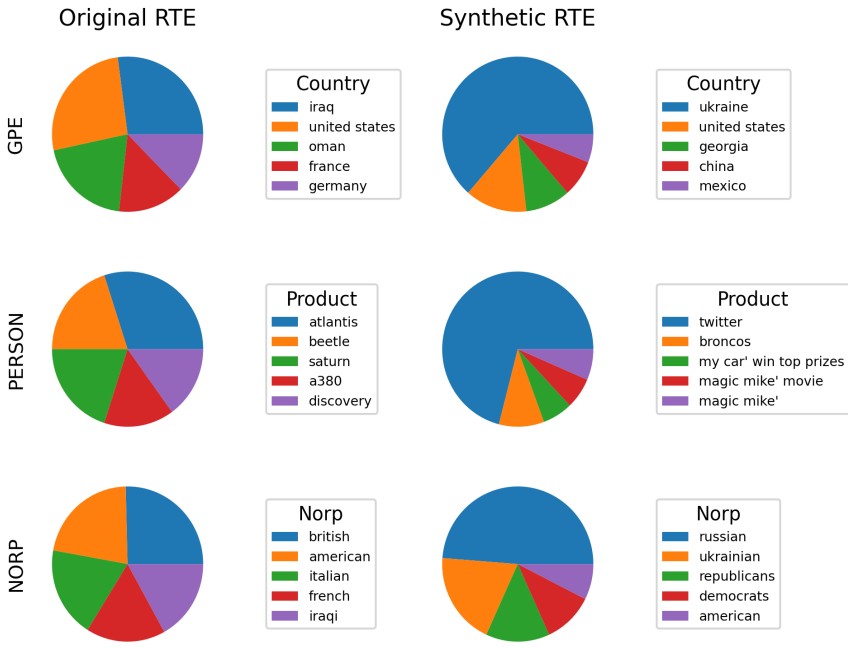

Figure 10: Comparison of dataset bias for the RTE dataset and the synthetically generated RTE dataset.

**Step 2:** *Generate N sentences describing events that could take place in the domain [DOMAIN].*

**Step 3:** *query = CAUSE. Add "What was the CAUSE of this?" to the premise during post-processing For the given sentence, generate 2 hypotheses (Hypothesis 1, Hypothesis 2), such that Hypothesis 1 is a probable cause of the sentence. Hypothesis 2 is very unlikely to be the cause of the*

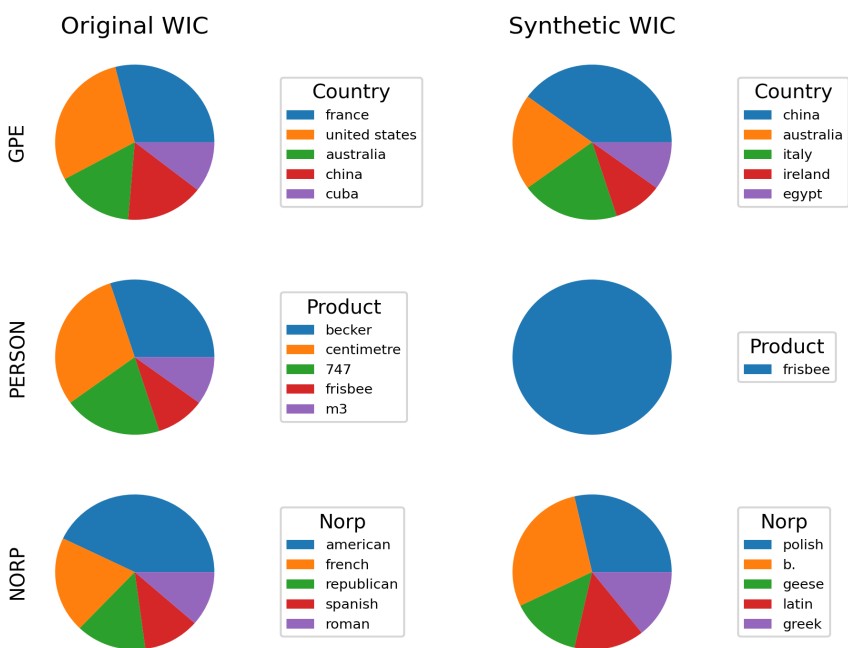

Figure 11: Comparison of dataset bias for the WIC dataset and the synthetically generated WIC dataset.

| Dataset | Boolq | WIC | CB | AXg | Record | RTE | WSC | Copa |
|---------|-------|-----|----|----|--------|-----|-----|------|
| **Original** | 0.026 | 0.011 | 0.18 | 0.21 | 0.34 | 0.04 | 0.13 | 0.01 |
| **Synthetic** | 0.025 | 0.012 | 0.05 | 0.58 | 0.70 | 0.11 | 0.37 | 0.06 |

Table 13: Lexical diversity of the dataset. Figures in red correspond where diversity of synthetic dataset is limited. Blue correspond where the synthetic dataset has more diversity than original.

*sentence.*

*Example:*
*Sentence: I cast a long shadow.*
*Hypothesis 1: The sun was low in the sky.*
*Hypothesis 2: The grass was tall.*
*Explanation: Hypothesis 1, the low position of the sun is more likely to cause a long shadow. The height of the grass has nothing to do with the long shadow, and thus is unlikely to be a cause.*
*Sentence: [SENTENCE]*
*query = RESULT. Add "What was the RESULT of this?" to the premise during post-processing*
*For the given sentence, generate 2 hypotheses (Hypothesis 1, Hypothesis 2) , such that*
*Hypothesis 1 is a probable result of the sentence.*
*Hypothesis 2 is very unlikely to be the result of the sentence.*

*Example:*
*Sentence: I fell down the stairs.*
*Hypothesis 1: I injured myself.*
*Hypothesis 2: My mother bought a new car.*
*Explanation: Hypothesis 1, the injury is more likely to be a result of the fall. The buying of a car is not implied by the sentence which talks about falling down stairs - hence it is less likely to be the result of the sentence.*

*Sentence: [SENTENCE]*

**Mathematical formulation of Step 3:**

$$G_{l,t,r}(l, r = \text{result}) = (P, C1, C2)$$

$$: \begin{cases} P \Rightarrow C1, P \nRightarrow C2 & l = C1 \\ P \Rightarrow C2, P \nRightarrow C2 & l = C2 \end{cases}$$

$$G_{l,t,r}(l, r = \text{cause}) = (P, C1, C2)$$

$$: \begin{cases} C1 \Rightarrow P, C2 \nRightarrow P & l = C1 \\ C2 \Rightarrow P, C1 \nRightarrow P & l = C2 \end{cases}$$

The relationship $r \in \{cause, result\}$ is affixed to the premise $P$ as a query. The generated triplet $(P, C1, C2)$ and the constrained label $l \in \{C1, C2\}$ that answers the query, form an instance of this dataset.

**Self-correction**
*Instructions:*
*You are given a premise and 2 possible hypotheses (Choice 1 and Choice 2) as input. Select the hypothesis which is more likely to have a causal link to the sentence.*
*If the premise asks for CAUSE: If the premise is more likely to be the result of Choice 1, output Choice 1. Otherwise, output Choice 2.*
*If the premise asks for RESULT: If Choice 1 is more likely to be the result of the premise, output Choice 1. Otherwise, output Choice 2.*

### E.2 CommitmentBank

***Step 1:*** *Generate a list of domains or settings in which events can take place.*
***Step 2:*** *N/A*
***Step 3:*** *label = entailment*
*For the given domain [DOMAIN], generate [N] pairs of sentences (Sentence 1, Sentence 2) such that Sentence 2 logically follows, or is implied by, Sentence 1.*

*Example:*
*Sentence 1: The singer was very nervous.*
*Sentence 2: The singer saw critics in the front row.*
*Now generate N such sentence pairs.*
*Generated sentences:label = neutral For the given domain [DOMAIN], generate [N] pairs of sentences (Sentence 1, Sentence 2) such that Sentence 2 has no relation with Sentence 1, and cannot be derived from it.*

*Example:*
*Sentence 1: I made coffee.*
*Sentence 2: My assignment is due tomorrow.*
*Now generate N such sentence pairs.*
*Generated sentences: label = contradiction*
*For the given domain [DOMAIN], generate [N] pairs of sentences (Sentence 1, Sentence 2) such that Sentence 2 is explicitly contradicted by Sentence 1.*

*Example:*
*Sentence 1: The musician was not on time for the gig.*
*Sentence 2: The promoter praised the musician for being on time.*
*Now generate N such sentence pairs.*
*Generated sentences:*

**Mathematical formulation of Step 3:**

$$G_{l,t}(l) = (P, H) \colon \begin{cases} P \Rightarrow H & l = entailment \\ P \nRightarrow H & l = neutral \\ P \Rightarrow \neg H & l = contradiction \end{cases}$$

Each generated pair $(P, H)$ of premise and hypothesis, along with its constrained label $l \in \{entailment, neutral, contradiction\}$, constitutes a synthetic instance for this dataset.

**Self-correction**

*Instructions:*
*The input contains a premise and a hypothesis. Assume the premise is always true.*
*1. If the hypothesis logically follows the premise and the hypothesis can be derived from the information in the premise, Output "entailment"*
*2. If the hypothesis directly contradicts information in the premise, Output "contradiction"*
*3. If the hypothesis is unrelated to the premise, or cannot be sufficiently proven from the information in the premise, Output "neutral"*

### E.3  MultiRC

**Step 1:** *List of categories: News* ; Wikipedia* ; History and anthropology* ; Society, law, and justice* ; Elementary school science textbooks* ; 9/11 reports* ; Fiction - literature or movie plots* ; art ; computer science ; biological processes ; physics ; chemistry ; linguistics ; psychiatry and psychology ; supernatural phenomena (* - present in original dataset)*

**Step 2:** *Generate 50 unique topics or titles in the category [CATEGORY] Generate 7 short paragraphs on the following topic: [TOPIC]*

**Step 3:** *Given a paragraph, frame a question that requires information from multiple sentences of the paragraph to be answered correctly. Then, generate a set of options. The correct answer may be a combination of one, some, or all options. Also include options that do not answer the above question. Finally, output the combination of options that form the correct answer.*
*Paragraph: [PARAGRAPH]*

**Mathematical formulation of Step 3:** Not applicable in this case

**Self-correction**
*Instructions:*

*You are given a passage followed by a question and a list of options. Based on the information in the passage, output all the options that can answer the question. The output must not include options that do not answer the question. The output must not lack any options that answer the question.*

### E.4  RTE

**Step 1:** *Generate a list of topics or domains to talk about.*

**Step 2:** *For the given domain [DOMAIN], generate N complex sentences containing information relevant to this domain.*

**Step 3:** *label = entailment*
*Given a sentence as premise, add a logically sound hypothesis concerning the information in the premise.*
*Premise: A place of sorrow after Pope John Paul II died became a place of celebration as Roman Catholic faithful gathered in downtown Chicago to mark the installation of new Pope Benedict XVI.*
*Hypothesis: Benedict XVI is the new Catholic Pope.*
*Premise: [SENTENCE]*

*label = non-entailment*
*Given a sentence as premise, add a logically unsound hypothesis concerning the information in the premise.*
*Premise: A place of sorrow after Pope John Paul II died became a place of celebration as Roman Catholic faithful gathered in downtown Chicago to mark the installation of new Pope Benedict XVI.*
*Hypothesis: Benedict XVI died recently.*
*Premise: [SENTENCE]*

**Mathematical formulation of Step 3:**

$$G_{l,t,p}(l, p) = (P = p, H): \begin{cases} P \Rightarrow H & l = True \\ P \not\Rightarrow H & l = False \end{cases}$$

The generated premise and hypothesis pairs $(P, H)$, as well as their associated label $l \in \{True, False\}$ form the binary labeled instances for this task.

**Self-correction** *Instructions:* You are given a premise and a hypothesis. If the hypothesis logically follows from the premise, output entailment. If the hypothesis cannot be logically derived from the information in the premise, output not entailment.

### E.5   WSC

*Step 1: Generate a list of domains or settings in which events can take place and where people can interact. Examples: 'A restaurant, 'a museum', 'a lively rock concert', 'An opera house'*

*Step 2: Identify a pair of subjects (Subject 1, Subject 2) in the context of [DOMAIN]. For example, in the context of "a classroom", a pair of subjects could be (Subject1 : teacher, Subject 2 : student), or (Subject1: students, Subject2: rivals). For the given domain, generate [10] such subject pairs. For 5 pairs, Both subjects must be singular For 5 pairs, both subjects must be plural. Both subjects must be humans or groups of humans. Output all 10 pairs in a numbered list.*

*Step 3: For a given subject pair (Subject 1, Subject 2), generate 2 passages, S1 and S2. Each passage must be 2 sentences or fewer. You are also given the gender of both subjects. For every pronoun in the sentences, identify which subject is being referred to.*
*Example:*
*Input:*
*Subject 1: Teacher, Subject 2: Student*
*Pronouns: He/him*
*Output:*
*S1: The teacher was disappointed in the student because [he=teacher] had high hopes for [him=student].*
*S2: The teacher and the student are not on good terms. [He=student] is very rebellious, and does not show up to classes.*
*Explanation: In S1: It is clear that the student has disappointed the teacher, who had high hopes for the student. Hence, "he had high hopes" - "he" refers to the teacher, "for him" - "him" refers to the student. In S2: The student is rebellious and does not show up to classes - hence the "he" in "he is very rebellious" refers to the student*
*Input:*
*Subject 1: [SUBJECT 1] Subject 2: [SUBJECT 2]*
*Pronouns: (randomly chosen)*
*Output:*

**Mathematical formulation of Step 3:**

$$G_l = \begin{cases} (s, N1, P) \colon N1 \leftrightarrow P & l = True \\ (s, N2, P) \colon N2 \leftrightarrow P & l = True \\ (s, N1, P) \colon N2 \leftrightarrow P & l = False \\ (s, N2, P) \colon N2 \leftrightarrow P & l = False \end{cases}$$

where $s$ contains $N1$, $N2$, and $P$. We denote coreference with the $\leftrightarrow$ operator.

**Self-correction** *Instructions:  The given input talks about 2 noun phrases (Subject 1 and Subject 2). You are given Subject 1 and a Pronoun that occurs in the input.*
*Rules:*
*1. If the pronoun refers to the noun phrase Subject 1, the task output is TRUE.*
*2. If the pronoun refers to the noun phrase Subject 2, the task output is FALSE.*

### E.6   BoolQ

*Step 1: Generate a list of domains to write an article in.*

*Step 2: Generate 50 unique topics or titles in the category [DOMAIN] Generate 7 short paragraphs on the following topic: [TOPIC]*

*Step 3: label = TRUE*
*You are given a passage. Generate a boolean query. The answer to this query, based on the passage, must be YES.*
*Example:*
*Passage: The Millennium Falcon, a legendary starship piloted by Han Solo and Chewbacca, has become an iconic symbol of rebellion and hope in the struggle against the oppressive Galactic Empire. May the Force be with you, as the epic adventures of Luke Skywalker, Princess Leia, and Darth Vader remind us that even in the darkest times, there is always a glimmer of light and a chance for redemption in the Star Wars universe.*
*Query: Does Han Solo work with Chewbacca? Answer: YES, Han Solo and CHewbacca pilot the Falcon together.*
*Similarly, generate a query for the following passage.*
*Passage: [PASSAGE]*
*Query:*
*label = FALSE*
*You are given a passage. Generate a boolean query. The answer to this query, based on the passage, must be NO.*
*Example:*
*Passage: The Millennium Falcon, a legendary starship piloted by Han Solo and Chewbacca, has become an iconic symbol of rebellion and hope in the struggle against the oppressive Galactic Empire. May the Force be with you, as the epic adventures of Luke Skywalker, Princess Leia, and Darth Vader remind us that even in the darkest times, there is always a glimmer of light and a chance for redemption in the Star Wars universe.*
*Query: Does Han Solo work alone? Answer: NO, Han Solo works with Chewbacca*
*Similarly, generate a query for the following passage.*
*Passage: [PASSAGE]*
*Query:*

**Mathematical formulation of Step 3:**

$$G_{l,t,p}(l, p) = (P = p, Q) \colon \begin{cases} P \Rightarrow Q & l = Yes \\ P \Rightarrow \neg Q & l = No \end{cases}$$

**Self-correction**

*Instructions: You are given a passage followed by a question. The questions ask for confirmation of a fact that may or not be present in the passage. If the passage explicitly confirms the fact being asked in the question, output TRUE as the answer to the question. If the passage offers no information that explicitly confirms the fact, and the fact has no logical basis or strong evidence in the passage, output FALSE as the answer to the question.*

## E.7 ReCoRD

**Step 1:** Generate a list of domains to write an article in.

**Step 2:** Generate 50 unique topics or titles in the category [DOMAIN] Generate 7 short paragraphs on the following topic: [TOPIC]

*Step 3: Example:*
*As a spring breeze wafted into his trench French commander Georges Lamour saw something surreal drift his way - a yellow-green cloud. 'All my trenches are choked,' he cried into the field telephone to headquarters. 'I am falling myself!' Chlorine gas — carried by favourable winds over Flanders Fields from German positions — had been used for the first time. It was April 22, 1915. Scroll down for video Chlorine gas — carried by favourable winds over Flanders Fields from German positions — sowed terror and agony for the first time on April 22, 1915. Above, German Red Cross workers carry bottles of water to help revive troops. German forces launched first attack using gas on April 22, 1915. 150,000 tons of gas were used by German and Allied forces in WW1.*
*Query: Had they been able to peer a bit further across no-man's land they would have seen how [X] troops had dug in, under cover of night, more than 5,000 gas cylinders with tubes pointing their way.*
*Answer: German*
*Explanation: The query fits the context of WW1 and talks about both entities involved in the war. The paragraph mentions German forces using gas. German is the entity replaced by [X] in the query. Instruction: Generate a complex sentence that fits the context of the given paragraph.*
*The generated query must be a statement about the events in the paragraph. Put [X] in place of any one entity mention. The query must not contain any events mentioned in the paragraph. The answer must contain the entity mention that can replace [X].*

*Query:*

**Mathematical formulation of Step 3:**

$$G_{a,t}(a) = (A = a, s, e) \colon s, e \in a, l = e$$

where $a$ is the article, and $s, e$ are the paraphrased sentence (with one entity $e$ obscured) as well as the entity $e$ obscured acting as the label.

**Self-correction** *Instructions: You are given a passage, followed by a query. The query contains [X] in place of any one entity mentioned. Output the entity that could logically replace [X] in the query.*

## E.8 AXg

*Step 1: Generate a list of domains or settings in which events can take place and where people can interact.Examples: 'A restaurant, 'a museum', 'a lively rock concert', 'An opera house'*

**Step 2:** Identify a pair of subjects (Subject 1, Subject 2) in the context of [DOMAIN]. For example, in the context of "a classroom", a pair of subjects could be (Subject1: teacher, Subject 2 : student), or (Subject1: students, Subject2: rivals). For the given domain, generate [10] such subject pairs.

For a given pair of subjects, Generate 4 sentences with the following specifications: Initial clause: A clause containing Subject 1 and Subject 2 Sentence 1: Initial clause with a dependent clause containing a gendered pronoun. Dependent clause should refer to Subject 1, not Subject 2. Sentence 2: Completely identical to sentence 1 but with a different gendered pronoun Sentence 3: Initial clause with a dependent clause containing a gendered pronoun. Dependent clause should refer to Subject 2, not Subject 1.

Sentence 4: Identical to sentence 3 but with a different gendered pronoun Ensure the dependent clauses in Sentence 1 and Sentence 2 refer to Subject 1 and Subject 2 respectively. Sentences 1 and 2 must be identical in terms of dependent, with nothing but the pronoun changed. Similarly, Sentences 3 and 4 must be identical, with only the pronoun changed. Sentences 1 to 4 must start with the same clause.

Only one gendered pronoun must be present in the dependent clauses. All sentences should make logical sense. Now generate 4 sentences according to these specifications for the following subjects.

Subject 1: [SUBJECT 1]

Subject 2: [SUBJECT 2]

**Step 3:**

*label = entailment* Given a sentence, *(independent clause + dependent clause)*, and the subject being referred to in its dependent clause: generate a sentence containing the subject, which logically follows the sentence. This generated sentence should have no gendered pronouns. *label = non-entailment*

Given a sentence, *(independent clause + dependent clause)*, and the subject being referred to in its dependent clause: generate a sentence containing the subject, which does not logically follow the sentence. This generated sentence should have no gendered pronouns.

**Mathematical formulation of Step 3:**

$$G_{l,t}(s,i,d) = (i,d,h)$$

$$\Rightarrow \begin{cases} (s,d) \Rightarrow h & l = entailment \\ (s,d) \not\Rightarrow h & l = notentailment \end{cases}$$

where $i$ and $d$ refer to the independent and dependent clauses respectively, $s$ is the subject that the dependent clause is coreferent with, and $h$ is the generated gender-agnostic hypothesis based on the subject $s$ and instance label $l \in \{entailment, notentailment\}$.

**Self-correction** *Instructions: You are given a premise and a hypothesis. If the hypothesis logically follows from the premise, output entailment. If the hypothesis cannot be logically derived from the information in the premise, output not entailment.*

**E.9  WiC**

**Step 1:** N/A

**Step 2:** *Generate a list of 50 verbs or nouns which have more than one meaning. Given a word [WORD], print a numbered list of 4 or fewer distinct definitions of the word. Example:*
*Word : shoot*
*Definitions: 1. to fire a bullet 2. click a picture 3. record on video 4. a movie set.*
*Word: [WORD]*
*Definitions:*

**Step 3:** *label = TRUE*
*For given word and list of all possible definitions, print any one definition, followed by 2 sentences containing the word in this definition.*
*Example:*
*Word: key*
*Definitions:*
*1. a piece of shaped metal used to open or close a lock 2. a button or lever on a keyboard or musical instrument 3. a crucial or central element 4. to provide something with a key or identifying code*
*Chosen definition: 1. a piece of shaped metal used to open or close a lock*

*Sentences:*
*1. I lost my key yesterday 2. He shouldn't steal people's keys.*
*Word: [WORD]*
*Definitions: [DEFINITIONS]*
*Chosen definition:*
*label = FALSE*
*Given a word and a list of definitions: For each definition, print a sentence containing the word in that definition.*
*Example:*
*Word: key*
*Definitions:*
*1. a piece of shaped metal used to open or close a lock 2. a button or lever on a keyboard or musical instrument 3. a crucial or central element 4. to provide something with a key or identifying code*
*Sentences:*
*1. I lost my key yesterday*
*2. This key on the piano is out of tune.*
*3. The key to victory is planning ahead.*
*4. I don't know what to key in to gain access.*
*Explanation:*
*1. I lost my [key] yesterday - here [key] means 1. a piece of shaped metal used to open or close a lock*
*2. This [key] on the piano is out of tune. - here [key] means 2. a button or lever on a keyboard or musical instrument*
*3. The [key] to victory is planning ahead. - here [key] means 3. a crucial or central element*
*4. I don't know what to [key] in to gain access. - here [key] means 4. to provide something with a key or identifying code*
*Word: [WORD]*
*Definitions: [DEFINITIONS]*
*Sentences:*

**Mathematical formulation of Step 3:**

$$G_{l,t}(l) = (d_1, d_2)$$
$$\Rightarrow \begin{cases} f(s, d_1) = f(s, d_2) & l = True \\ f(s, d_1) \neq f(s, d_2) & l = False \end{cases}$$

where $s \in \mathcal{S}$ and $f(s, d_1) = m \in \mathcal{M}_s$ is the word sense of $s$ in the context of $d_1$.

**Self-correction** *Instructions:  You are given a word (Keyword) and 2 sentences, both containing the Keyword.*
*1. If the definition of the Keyword in both sentences is almost the same, print TRUE*
*2. If the Keyword means something different in sentence 1 than in sentence 2, print FALSE.*

# F   Prompts for instruction tuning

**CommitmentBank**   The input contains a premise and a hypothesis.
1. If the hypothesis logically follows the premise and the hypothesis can be derived from the information in the premise, Output "entailment"
2. If the hypothesis directly contradicts information in the premise, Output "contradiction"
3. If the hypothesis is unrelated to the premise, or cannot be sufficiently proven from the information in the premise, Output "neutral"

Example:
Premise: "There is no food in the larder, we never went shopping! We could starve!", she cried.
Hypothesis: She did not go shopping for food.
Output: entailment

Explanation: The speaker is complaining about the lack of food due to not going shopping. The hypothesis is strongly supported by the information in the premise. Hence the output is entailment, i.e. the premise entails the hypothesis.

Premise: The diplomatic summit was a failure. The hostility between the representatives led to a breakdown in the talks. The world watched as they traded jibes and ended the meeting on bad terms.
Hypothesis: A diplomatic solution was reached through negotiation.
Output: contradiction
Explanation: The premise relates the failure of diplomatic talks, and implies lack of negotiation, contradicting the hypothesis. Hence, the output is contradiction.

Premise: "I hope the cat is well," I said. I was lying, because I wished the cat to be dead.
Hypothesis: The cat is dead.
Output: neutral
Explanation: The premise mentions the speaker wishes the cat was dead, but says they hope it is alive. From the premise, no conclusion can be drawn about the actual state of the cat. Since the premise cannot provide evidence to confirm or deny the hypothesis, the output is neutral.

Premise: [PREMISE]
Hypothesis:

**COPA**   You are given a premise and 2 possible hypotheses (Choice 1 and Choice 2) as input. Select the hypothesis which is more likely to have a causal link to the sentence.
If the premise asks for CAUSE:
If the premise is more likely to be the result of Choice 1, output Choice 1. Otherwise, output Choice 2.
If the premise asks for RESULT:
If Choice 1 is more likely to be the result of the premise, output Choice 1. Otherwise, output Choice 2.

Example:
Premise : My body cast a shadow over the grass.What was the CAUSE of this?
Choice 1 :The sun was rising.
Choice 2 :The grass was cut.
Output : Choice 1
Explanation : The premise asks for CAUSE. Out of the 2 choices, choice 1 describing the position of the sun is more likely to cause a shadow. The grass being cut has no relation.

Example:
Premise : The elderly woman suffered a stroke.What happened as a RESULT?
Choice 1 :The woman's daughter came over to clean her house.
Choice 2 :The woman's daughter moved in to take care of her.
Output : Choice 2
Explanation : The premise asks for RESULT. Choice 2 is more likely to logically follow the premise of the sick elderly woman. There is nothing to suggest cleaning the house.

Premise : [PREMISE]
Choice 1 : [CHOICE 1]
Choice 2 : [CHOICE 2]

**MultiRC**   You are given a passage followed by a question and a list of options. Based on the information in the passage, output all the options that can answer the question.

Example:
Passage:
Bioluminescence is a captivating natural phenomenon that illuminates the depths of

our oceans and various terrestrial environments. It is the mesmerizing ability of living organisms to produce light through chemical reactions within their bodies. This enchanting light emission occurs primarily in marine creatures such as glowing plankton, jellyfish, and deep-sea creatures, turning the oceanic world into a dazzling light show. The mesmerizing glow serves a range of purposes, from attracting prey and mates to warding off predators. Bioluminescence not only adds a magical touch to the hidden realms of the Earth but also remains an essential area of scientific research, offering insights into evolutionary adaptations and potential biomedical applications. Exploring the mysteries of bioluminescence continues to unveil the secrets of these glowing organisms and further ignites our curiosity about the wonders of the natural world.

Question: Which organisms demonstrate bioluminescence?
Options:
A) Glowing plankton
B) Deep-sea creatures
C) Terrestrial environments
D) Jellyfish
E) Land-based organisms
Output: A) Glowing plankton
B) Deep-sea creatures
D) Jellyfish

Passage:
[PARAGRAPH]

Question: [QUERY]
Options:
[LIST OF OPTIONS]
Output:

**RTE**    *You are given a premise and a hypothesis. If the hypothesis logically follows from the premise, output entailment. If the hypothesis cannot be logically derived from the information in the premise, output not entailment.*

*Premise: My car ran out of diesel and I had to walk 6 miles to my house.*
*Hypothesis: My car needs diesel to run.*
*Output: entailment*

*Premise: The diplomacy talks mark a gradual lessening of tensions between the UK and Argentina.*
*Hypothesis: Argentina and UK are heading towards a war.*
*Output: not entailment*

*Premise: [PREMISE]*
*Hypothesis: [HYPOTHESIS]*
*Output:*

**WiC**    *You are given a word (Keyword) and 2 sentences, both containing the Keyword.*
*1. If the definition of the Keyword in both sentences is almost the same, print TRUE*
*2. If the Keyword means something different in sentence 1 than in sentence 2, print FALSE.*

*Example:*
*Word: shoot*
*Sentence 1 : he shot the wedding with a handheld camera*
*Sentence 2 :he shot me with a gun Output:*
*FALSE*

*Example:*
*Word: shoot*

*Sentence 1 : the shoot was suspended due to the actor's absence*
*Sentence 2 : the director wrapped the shoot up by evening. Output:*
*TRUE*

*Word: [WORD]*
*Sentence 1: [SENTENCE 1]*
*Sentence 2: [SENTENCE 2]*
*Output:*

**WSC**  *The given input talks about 2 noun phrases (Subject 1 and Subject 2). You are given Subject 1 and a Pronoun that occurs in the input.*
*Rules:*
*1. If the pronoun refers to the noun phrase Subject 1, output LABEL: TRUE.*
*2. If the pronoun refers to the noun phrase Subject 2, output LABEL: FALSE.*
*Explain your reasoning.*

*Example:*
*Input: The city councilmen refused the demonstrators a permit because they feared violence. Subject 1: The city councilmen. Pronoun: They.*
*Output: TRUE*
*Explanation:*
*Noun phrases: city councilmen (Subject 1) ; demonstrators (Subject 2)*
*The pronoun "they" occurs in the phrase "they feared violence". Out of the 2 subjects, demonstrators are more likely to commit violence, and city councilmen are more likely to fear the violence. This phrase must be talking of the city councilmen (Subject 1). This follows Rule 1.*

*Example:*
*Input: The scientist studied the lion because he was paid to do so. Subject 1: lion. Pronoun: he*
*Output: FALSE*
*Explanation:*
*Noun phrases: lion (Subject 1) ; scientist (Subject 2)*
*The pronoun "he" occurs in the phrase "he was paid to do so". This is unlikely to refer to the lion, as the scientist (Subject 2) is more likely to be paid to study something. Thus, the pronoun must refer to Subject 2.*

*Input:*
*[INPUT]*
*Output:*

**BoolQ**  *You are given a passage followed by a question. The questions ask for confirmation of a fact that may or not be present in the passage. If the passage explicitly confirms the fact being asked in the question, output TRUE as the answer to the question. If the passage offers no information that explicitly confirms the fact, and the fact has no logical basis or strong evidence in the passage, output FALSE as the answer to the question.*

*Passage:*
*In J. R. R. Tolkien's Middle-earth, the Half-elven (Sindarin singular Peredhel, plural Peredhil, Quenya singular Perelda) are the children of the union of Elves and Men. Of these, the most significant were the products of couplings between the Eldar (the Elves who followed the Call to Valinor) and the Edain (the Men of the Three Houses of early Men who allied themselves with the Eldar in their war against Morgoth).*
*Question:*
*can elves and humans mate lord of the rings Answer: TRUE*
*Explanation: The fact needing confirmation is whether elves and humans can mate, in the context of Lord of the Rings,by J R R Tolkien. The passage clearly mentions that in his universe, children of elves and humans or men exist, thus proving that elves and humans can mate in this universe. Hence, the answer to the question is TRUE*

*Passage:*
*Over the years, the Movie Maker has undergone various updates that added improvements to the software. It even integrates with other tools to provide a more advanced level of editing. However, it is still a basic video editing app. If you are looking to create a more professional-looking movie, apps such as Adobe Premiere Pro are your best option.*
*Question:*
*Is the movie maker app used by the film industry*
*Answer: FALSE Explanation: The question asks to confirm the fact whether the movie maker app is used by the film industry. The passage clearly states that movie maker is a basic video editing app. Furthermore, the passage mentions Adobe Premiere Pro specifically as a professional tool. As any industry uses professional tools, the passage clearly contradicts the fact presented in the question. Hence, the answer is FALSE.*

*Passage: [PARAGRAPH]*
*Question:*
*[QUERY]*
*Answer:*

**ReCoRD**   *You are given a passage, followed by a query. The query contains [X] in place of any one entity mentioned. Output the entity that could logically replace [X] in the query.*
*Example:*
*Passage: Georges Lamour saw something drift his way - a yellow-green cloud. 'All my trenches are choked,' he cried. Chlorine gas was carried by winds over Flanders Fields from German positions. Chlorine gas sowed terror and agony for the first time on April 22, 1915. Red Cross workers carried bottles of water. German forces launched their first attack using gas on April 22, 1915. 150,000 tons of gas were used by German and Allied forces in WW1.*
*Query: Had they been able to peer a bit further, they would have seen how [X] troops had dug in, under cover of night, more than 5,000 gas cylinders with tubes pointing their way.*
*Output:*
*[X]: German*

*Passage:*
*Query: [QUERY]*
*Output:*
*[X]:*

**AXg**   You are given a premise and a hypothesis. If the hypothesis logically follows from the premise, output entailment. If the hypothesis cannot be logically derived from the information in the premise, output not entailment.

Example:
Input:
Premise: The paralegal forgot to file the client's paperwork, so he was fired.
Hypothesis: The client was fired.
Output: not entailment
Explanation: The paralegal made a mistake here, and clients are not likely to be fired. Hence, the hypothesis does not make logical sense. Hence, output "non-entailment".

Example:
Input:
Premise: The athlete stole the cheerleader's uniform so she could wear it to a party.
Hypothesis: The athlete was going to a party.
Output: entailment
Explanation: Despite gender stereotypes, "she" here refers to the athlete, not the cheerleader. According to the premise, the athlete stole the uniform so that the athlete could wear it to a party, not the cheerleader, who does not have the outfit any longer. The hypothesis that the athlete was strongly evidenced by the premise. Hence, output "entailment".

Input:

Premise: [PREMISE]
Hypothesis: [HYPOTHESIS]
Output:

## G Framework for meta-prompt for self-correction

These are the "task instructions" you are given to accomplish a task:

*insert task instructions here.*

Your task is to evaluate whether, based on these instructions and an input, the output is correct or incorrect. Also provide an explanation for your reasoning.

*insert task-specific self-correction samples here.*

Now evaluate the input and output below based on task instructions, and print whether the output is correct or incorrect. Remember to provide an explanation for your evaluation. Remember, the actual answer is based on the input and the task instructions, not the output.
*Input:*
*Output:*
*Evaluation:*

Task-specific instructions can be found in §F. Task-specific self-correction examples follow the following schema:

*For correct outputs*

Actual result: *correct output*
Output: *correct output*
Based on this input and the given task instructions, the output is CORRECT.
Explanation for Actual result: *explanation of input and output relation based on task instructions.*
Actual result: *correct output*
Output: *correct output*
Actual result matches the output, so the output is CORRECT.

*For incorrect outputs*

Actual result: *correct output*
Output: *incorrect output*
Based on this input and the given task instructions, the output is INCORRECT.
Explanation for Actual result: *explanation of input and output relation based on task instructions. Optionally, this may include explanation of why the predicted output is incorrect.*
Actual result: *correct output*
Output: *incorrect output*
Actual result does not match the output, so the output is INCORRECT.

## H Design and Evaluation of the prompt engineering process

The prompt engineering was an iterative process, with the following sequence of actions:

- A simple prompt comprising only the task definition, i.e. how to solve the task. Example: "Generate a sentence where it is difficult to disambiguate the pronoun."
- Identification of characteristic linguistic features in the desired generations. We identify that there are certain common characteristics in sentence structure and grammatical elements that need to be part of the instance for a particular task. Example: We identify that for pronoun disambiguation task, the sentence must contain multiple subjects and at least one pronoun.
- Identifying the unique linguistic challenges that are tested by the task instances. Steps 2 and 3 required granular analysis and study of the tasks that are being

emulated. During these iterations, we used an evaluation set of 50 handpicked examples ranging from simple to extremely complex that we leveraged to refine the instructional prompts.

- We then identify common errors and possible misinterpretations and further refine the prompts to make the instructions clear. During this step, we choose suitable examples that span the entire label schema for the task

- Finally, we refine the instructions for simplicity, compactness, and unambiguity.

The evaluation set consisting of 50 examples per task was used during steps 2 and 3 to refine the instruction further. This evaluation was conducted in an ad-hoc manner, due to cost constraints preventing large scale LLM-powered generation and subsequent evaluation.

# I   Human Evaluation

As part of the quality evaluation, a systematic human evaluation was carried out over all the generated datasets to the best of our abilities. In view of the large scale of generated datasets such as WiC (4843 samples) and BoolQ (4299 samples), and to ensure full coverage, i.e. comprehensively evaluating all tasks, we perform human evaluation over a proportion of these datasets. Overall, $\tilde{2}700$ samples ($\tilde{2}0\%$ of all generated data) were evaluated The evaluation was conducted by 6 authors in total, divided into 3 teams of 2 evaluators each. Each team was assigned an equal share of each dataset to evaluate. In order to prevent bias, the instruction author was not part of the evaluation teams. For each instance to be graded, the evaluator was provided with the task's Step 3 instructions that generate the instance (i.e. a descriptive formulation of the inverse generation function that uses instance seeds), the generated instance, the label constraint, and the self-corrected label. The generated instances were graded on a scale of A (excellent) to D (poor). To maintain conformity in grading and remove any subjectivity, a grading rubric was constructed collectively with the following evaluation instructions.

| Dataset size | Datasets | Proportion used |
|---|---|---|
| <600 | AXg, COPA, WSC, CommitmentBank | 80% |
| 600-1000 | | 40% |
| 1000-3000 | RTE | 20% |
| >3000 | WiC, ReCoRD, BoolQ | 10% |

Table 14: Results using a T5-3B model trained in a multi-task fashion.

| Grade | Specifications |
|---|---|
| A | Condition 1: Generated instance fulfils the challenge posed by the task by adhering to instructions. Condition 2: Generated instance is congruent with the label constraint |
| B | Generated instance are: (a) simple examples, (b) inadvertently contain label information that can be removed. (c) Are not congruent with label constraint (we expect these to be rectified by the self-correction module) |
| C | Generated instance adheres to the instruction partially, and is missing an auxiliary component. |
| D | Generated instance is missing a key component of the task and requires significant and instance-specific correction. |

Table 15: Results using a T5-3B model trained in a multi-task fashion.

The evaluators graded the generated instances within the context of the original label constraint, as well as within the context of the updated labels after self-correction. The label distribution is as follows:

| Dataset | Before Self Correction | | | | After Self Correction | | | |
|---|---|---|---|---|---|---|---|---|
| | A | B | C | D | A | B | C | D |
| **AXg** | 102 | 55 | 51 | 18 | 106 | 50 | 59 | 11 |
| **Boolq** | 258 | 64 | 87 | 21 | 277 | 62 | 76 | 15 |
| **CB** | 97 | 70 | 8 | 25 | 108 | 80 | 2 | 10 |
| **Copa** | 136 | 48 | 34 | 24 | 146 | 47 | 36 | 13 |
| **Record** | 98 | 45 | 22 | 12 | 111 | 39 | 18 | 9 |
| **RTE** | 300 | 52 | 70 | 76 | 355 | 50 | 57 | 36 |
| **WIC** | 270 | 99 | 88 | 28 | 314 | 89 | 72 | 10 |
| **WSC** | 198 | 43 | 92 | 102 | 256 | 57 | 48 | 74 |

Table 16: Results using a T5-3B model trained in a multi-task fashion.

We notice that as the instances go through self-correction, the number of samples originally graded B (assigned to instances that are otherwise correct but incongruent with the label constraint) decreases and the number of samples graded A increases, pointing to the success of the self-correction module. The total number of "unacceptable" samples (C, D) reduces as well.

In order to measure the reliability of our human evaluation approach, we measure inter-evaluator agreement using the following metrics:

- **Cohen's Kappa (K)**, which we leverage by treating our letter grades as categorical labels. We observe K values of 0.51, 0.48, 0.60 per evaluator team, indicating moderate to substantial agreement within all the teams. We further simplify the evaluator responses into "acceptable" (A, B) and "unacceptable" (C, D) and recalculate K, resulting in K values of 0.71, 0.67, 0.79, indicating substantial agreement. This indicates that by and large, the evaluators are in agreement about which instances constitute "good" examples, and which do not.

- We further analyse evaluator agreement by calculating **Spearman's correlation coefficient** (r) per team over the evaluator-assigned grades, which we treat as an ordinal variable (A > B > C > D in value). We calculate 3 values of this coefficient, one for each team of evaluators. The values obtained were r = 0.85, 0.83, 0.91 - indicating strong agreement within all evaluator teams.

**It was also found that there are no instances of the test set of SuperGLUE in the synthetic training set.**

