# OpenReview forum: "TarGEN: Targeted Data Generation with Large Language Models"
_colmweb.org/COLM/2024/Conference — COLM_

### Official Review · Reviewer_oq9P · 2024-05-08

**Rating:** 6
**Confidence:** 3
**Ethics Flag:** 1

**Summary:**

The paper introduces TarGEN, a novel framework for generating high-quality, synthetic datasets using large language models (LLMs) without relying on existing data instances. TarGEN utilizes a multi-step prompting strategy that aims to inject semantic diversity and ensure precise label alignment through a label-constrained generation process. This strategy includes initializing contextual diversity, generating task-specific instance seeds, and employing a self-correction module to enhance label accuracy. The authors test TarGEN by emulating tasks from the SuperGLUE benchmark and training various models on these synthetic datasets. The results demonstrate that models trained on TarGEN-generated datasets generally outperform those trained on original datasets, with improvements noted across different model architectures.

**Reasons To Accept:**

1. The proposed TarGEN does not require pre-existing data for specific tasks, making it versatile for generating datasets across a wide range of domains and tasks, including novel or specialized tasks that lack substantial existing datasets. This flexibility is particularly valuable in the rapidly evolving field of AI, where new tasks and requirements emerge frequently.

2. The incorporation of a self-correction module within TarGEN reduces label noise, ensuring that the synthetic data is not only high in quantity but also in quality. This approach leads to more reliable training data, which is critical for developing robust and accurate models.

**Reasons To Reject:**

1. The performance improvements attributed to TarGEN could be heavily reliant on sophisticated, handcrafted prompt engineering. While the paper suggests that the framework is adaptable across tasks, the extent to which performance is dependent on the skillful design of prompts and the labor-intensive process of configuring these prompts for different tasks should be more transparently addressed.

2. The most significant concern is the lack of extensive comparative analysis with other existing algorithms and an ablation study of the 'label-constrained generation' facilitated by 'diverse seeds.' The paper would benefit from a deeper exploration of how each component of the TarGEN methodology contributes to the overall performance improvements.

---

> ### Author Rebuttal · Authors · 2024-05-31
>
> * **R4.1 - Sophisticated Prompt Engineering**: We thank the reviewer for the feedback. We employ it to generate samples for a sarcasm identification task:
>   - Step 1: Generate a list of domains or settings in which events can take place and where people can interact. Examples: ’A restaurant, ’a museum’, ’a lively rock concert’
> Response: “an opera house”. “a classroom”, “a wedding venue” ….
>   - Step 2: For all contexts | Prompt: Generate a list of conversational questions you might hear in a conversation in [an opera house].
> Response: Did you read the synopsis before coming?
>   - Step 3: Label = TRUE | Prompt: Generate a sarcastic response for the following question: Did you read the synopsis before coming?.
> Example response: Of course not, I love being completely lost for three hours.
> Label = FALSE | Prompt: Generate a genuine, non-sarcastic response for the following question: Did you read the synopsis before coming?
> Example response: Yes, I did. It helped me understand the context and enjoy the performance more.
>
> This demonstrates that we can use simple instructions within the TarGEN framework to generate a majority of language tasks.
>
> We would like to highlight that the simple nature of our framework allows for greater flexibility and control, making it well-suited to domain-specific or particularly complex tasks. We would also request the reviewer to go through our response **R1.1** addressing the same concern. We are happy to generate any other tasks that the reviewer deems necessary for the verification of this approach.
>
> * **R4.2.1 - Lack of comparative analysis**: Thanks for the excellent feedback. It has helped us improve our paper. We showed additional comparative analysis to highlight the differences between our work and previous data generation methods. Please check responses **R1.2** & **R2.2** where we present the qualitative and quantitative results as part of the additional experiments.
>
> * **R4.2.2 - Ablation studies**: We carry out additional experiments by ablating the "diverse seeds" component as well as the "generative model" in our pipeline. Please check responses **R2.3** and **R3.1** respectively for the same.
>
>
> We would be happy to answer additional concerns raised by the reviewer.

---

> > ### Author Response · Authors · 2024-06-03
> > **Follow up to the Rebuttal**
> >
> > Dear Reviewer,
> >
> > We trust this message finds you well. We are writing to follow up on the rebuttal response we submitted in response to your feedback. We are eager to receive your insights and comments on our response. If there are any areas that you still find unclear or require further clarification, we would be more than happy to provide additional information to ensure a comprehensive understanding of our work.
> >
> > Your feedback is of utmost importance to us, and we greatly appreciate your time and consideration in evaluating our research.
> >
> > Thank you for your attention, and we look forward to hearing from you soon.

---

> > > ### Author Response · Authors · 2024-06-06
> > > **Follow up to the Rebuttal**
> > >
> > > Dear Reviewer,
> > >
> > > We hope this email finds you well. We are following up on our rebuttal response to your valuable feedback on our research. As a reminder, the deadline for rebuttal responses is tomorrow, June 6th. We would be grateful if you could share any further insights or comments you may have on our response at your earliest convenience.
> > > We truly appreciate your time and thoughtful review. Your feedback is crucial to improving our work, and we value your expertise in this field.

---

> > ### Comment · Reviewer_oq9P · 2024-06-06
> > **Thanks for your responses**
> >
> > Thanks for your responses and new results. I've raised my scores.

---

### Official Review · Reviewer_b77X · 2024-05-10

**Rating:** 7
**Confidence:** 4
**Ethics Flag:** 1

**Summary:**

The authors propose TarGEN, a multi-step dataset generation technique based on CharGPT to generate synthetic labeled datasets that can be used to evaluate the performance of NLP models. They additionally propose a self-correction module that automatically evaluate the generated labels to ensure the quality of the generated dataset. To evaluate its performance, TarGEN was used to generate a synthetic SuperGLUE dataset where different NLP models were finetuned on the synthetic SuperGLUE and the original one. The results showed that models finetuned on the generated SuperGLUE outperformed models trained on the original SuperGLUE. TarGEN also showed to perform well when compared with other synthetic data generation methods such as Self-Instruct.

**Questions To Authors:**

- How do you ensure consistency when generating new data using TarGEN? For example, how can TarGEN be used to generate 1000 similar labeled samples across multiple separate runs (let's say 5 runs)?

- How do you enhance the diversity of the generated data?

- How do you ensure that TarGEN does not generate toxic sentences?

**Reasons To Accept:**

- The paper is well-written, easy to follow, and clear.
- Synthetic dataset generation is an active research that is shaping the way we evaluate the learning capabilities of LLMs. Therefore reduce the time and cost needed to manually generate new datasets.
- TarGEN can be used to generate datasets for novel domain-specific tasks with no existing instances.
- TarGEN has show to outperform other similar techniques such those used to generate Self-Instruct dataset.

**Reasons To Reject:**

- TarGEN only relied on ChatGPT and there was no comparison about how using a different LLM could affect the generated data.

---

> ### Author Rebuttal · Authors · 2024-05-30
>
> * **R3.1 - Comparison with other generative models**: We appreciate the feedback of the reviewer. We perform the data generation pipeline with Claude Sonnet and Llama 3 70B to create 2 more variants of synthetic SuperGLUE with each model keeping the task set, prompts, and pipeline steps constant. Llama 2 7b was fine-tuned on the above-mentioned datasets and evaluated on the original test set. The results of the experiments are attached below:
>
> | Train data generated by: | BoolQ | WiC   | CB    | AX-g  | ReCoRD | RTE   | WSC   | COPA  |
> | ------------------------ | ----- | ----- | ----- | ----- | ------ | ----- | ----- | ----- |
> | ChatGPT (current)        | 90.12 | 73.21 | 94.87 | 51.32 | 78.81  | 92.89 | 77.32 | 95.72 |
> | Claude Sonnet            | 90.65 | 73.88 | 95.23 | 51.88 | 79.22  | 93.01 | 77.89 | 95.72 |
> | Llama 3 70B              | 89.56 | 73.33 | 93.65 | 51.88 | 79.01  | 92.23 | 77.65 | 95.23 |
>
> We see nearly the same or slightly improved scores when using Llama 3 70B and Claude Sonnet. We would be happy to conduct additional experiments.
>
> * **R3.2 - Ensuring consistency**: We ensured data consistency by controlling the decoding parameter, temperature, presence penalty, and frequency penalty. Additionally, one can ensure consistency with a more strict constraint of keeping the generated initial seeds frozen to ensure data consistency across five runs.
>
> * **R3.3 - Enhancing diversity**: The first step of the pipeline allows us to generate multiple semantic contexts within which these instances occur. That in turn helps to generate seed instances with increased semantic diversity. To quantitatively show the effect of seed tasks, we experimented to see the impact if we reduced the number of semantic contexts. The results are mentioned in response **R2.3**.
>
> * **R3.4 - Controlling toxicity**: To demonstrate a task-agnostic framework, we rely on guardrails built into the publicly accessible inference interfaces of leading Language models like ChatGPT, Llama 3, and Claude. However, the framework is flexible to the addition of any safeguards that data generators may wish to include for their purposes.
>
> We would be happy to answer any further questions or perform additional experiments as required by the reviewer.

---

> > ### Author Response · Authors · 2024-06-03
> > **Follow up to the Rebuttal**
> >
> > Dear Reviewer,
> >
> > We trust this message finds you well. We are writing to follow up on the rebuttal response we submitted in response to your feedback. We are eager to receive your insights and comments on our response. If there are any areas that you still find unclear or require further clarification, we would be more than happy to provide additional information to ensure a comprehensive understanding of our work.
> >
> > Your feedback is of utmost importance to us, and we greatly appreciate your time and consideration in evaluating our research.
> >
> > Thank you for your attention, and we look forward to hearing from you soon.

---

> > ### Comment · Reviewer_b77X · 2024-06-04
> >
> > Thanks for the detailed feedback on my concerns! Congratulations on the great work!

---

### Official Review · Reviewer_QmfM · 2024-05-12

**Rating:** 6
**Confidence:** 3
**Ethics Flag:** 1

**Summary:**

This paper proposes a method for generating synthetic data by going through a number of steps such as task description, label-constrained generation, and self-correction.

**Questions To Authors:**

Question 1. Does ChatGPT refer to gpt-3.5-turbo? If so, please specify which model was used in the dataset generation with its model name (including version number) for better clarity and reproducibility purposes.

**Reasons To Accept:**

The strengths of the paper are as follows:

Strength 1: Overall the method seems reasonable, including several steps that would likely increase the quality of the data.

Strength 2: Experiments are performed over a variety of models.

Strength 3: I like the fact that human evaluation is done to verify the quality of the data.

**Reasons To Reject:**

But I had several concerns about the paper in its current form:

Weakness 1. The conceptual comparison with previous methods for dataset generation such as SuperGen, ZeroGen, SunGen, ProGen, and Self-Instruct is weak. It would be good to have the exact differences between these methods explained, e.g. in a table.

Weakness 2. Similarly, the empirical comparison to these methods is weak. There is only one comparison with Self-Instruct, and that is through tuning on the Self-Instruct dataset, which (as far as I know) was not specifically tailored to result in omdels that are good at the tasks in SuperGlue. A more comprehensive comparison would have used Self-Instruct and the other related methods above to generate data specifically for the tasks in the SuperGlue dataset (like is done for TarGen), and demonstrate the differences caused by the proposed methodology, while keeping the task set constant.

Weakness 3. There were not any ablations to elucidate the importance of each of the elements that went into TarGen.

Weakness 4. The discussion of the V-usable information was unclear because (a) it was not explained exactly how v-usable information was calculated in the paper, and (b) it seems that the v-usable information actually diverges significantly from that of the original data, which seems like it is more of a problem than a positive trait.

---

> ### Author Rebuttal · Authors · 2024-05-30
>
> * **R2.1 - Qualitative comparison with other methods**: Please check the response R1.2.
> * **R2.2 - Empirical comparison with other methods**: To show a quantitative comparison with other data generation methods, we generate synthetic versions of specific tasks from the SuperGLUE dataset using recent data generation frameworks:
> The tasks we generate are RTE (textual entailment) and AXg (gender disambiguation). We then train a Llama-2 model on these synthetic task variants and evaluate them on original test sets of the datasets. The results are as follows:
> | Method|RTE|AXg|
> |-|-|-|
> |CODA|64.56 |23.46|
> |ZeroGEN|58.54|19.01|
> |SuperGEN|77.94|44.67|
> |SunGEN|75.11|38.16|
> |ProGEN|78.43|37.94|
> |Ours (TarGEN)|92.89|51.32|
>
> * **R2.3 - Ablation studies**: Thank you for your valuable feedback. We have incorporated ablation studies to quantitatively show the effect of each component. For the seed instances, we experimented to see the impact of reducing the number of semantic contexts.
>
> |Scores with| BoolQ|WiC|CB|AX-g|ReCoRD|RTE|WSC|COPA|
> |-|-|-|-|-|-|-|-|-|
> |100%|90.12|73.21|94.87|51.32|78.81|92.89|77.32|95.72|
> |50%|85.43|72.65|90.33|49.86|75.42|89.75|76.21|92.11|
> |25%|83.65|71.78|88.47|48.32|73.78|87.81|76.02|91.21|
>
> We also showed the effect of self-correction as a part of the ablation study. We are happy to conduct more experiments as per the reviewers' suggestions.
>
> * **R2.4 - Regarding $\mathcal{V}\$-Usable Information**: $\mathcal{V}\$-usable information is calculated as: $I_{V}(X \rightarrow Y) = H_{V}(Y) - H_{V}(Y|X)$, where $H_{V}(Y)$ is the predictive entropy from a null model $LM_{\text{Null}}$ (trained on empty inputs), and $H_{V}(Y|X)$ is the conditional entropy from a model $LM_{X}$ (trained on input samples and labels). Entropies are measured in bits using $\log_2(P)$, where $P$ is the probability distribution of the labels assigned by the specific model. Higher $\mathcal{V}\$-usable information indicates harder samples, while lower values indicate easier samples for model family $\mathcal{V}\$. Original datasets have $\mathcal{V}\$-usable information concentrated in a narrow range of easier samples. Our strategy generates a broader range of $\mathcal{V}\$-usable information, from easy to hard, leading to better finetuning results.
>
> * **R2.5 - ChatGPT version clarification**: Yes, Version number is 0613. This information will be added in the camera ready version.

---

> > ### Author Response · Authors · 2024-06-03
> > **Follow up to the Rebuttal**
> >
> > Dear Reviewer,
> >
> > We trust this message finds you well. We are writing to follow up on the rebuttal response we submitted in response to your feedback. We are eager to receive your insights and comments on our response. If there are any areas that you still find unclear or require further clarification, we would be more than happy to provide additional information to ensure a comprehensive understanding of our work.
> >
> > Your feedback is of utmost importance to us, and we greatly appreciate your time and consideration in evaluating our research.
> >
> > Thank you for your attention, and we look forward to hearing from you soon.

---

> > > ### Comment · Reviewer_QmfM · 2024-06-04
> > > **Thank you for the response**
> > >
> > > Thank you for the response, and I think the resulting paper is significantly improved. I think a more comprehensive comparison with previous methods that further explains *why* TarGen outperforms them significantly on these tasks would further improve the paper. I have raised my rating to 6.

---

### Official Review · Reviewer_BhhP · 2024-05-18

**Rating:** 8
**Confidence:** 4
**Ethics Flag:** 1

**Summary:**

TarGen is a seed-less data augmentation method that uses multi-step prompting along with self-correction. The seed-less nature allows it to generate new data for tasks as opposed to augmenting from seed data for a task. The self-correction module is designed to correct errors such that generated sample is relevant to the given label

The steps in generation are broadly 1) Generating some task specific keywords to have a diverse set of 2) Generating some seed instances (these are still not the final input samples) 3) Generate input samples based on the seed instances and constrained by labels 4) Error correction to ensure label relevance and relabel if required

**Questions To Authors:**

If instructions are varied in a step, how does that affect the overall performance in the downstream task?

**Reasons To Accept:**

1) The approach is flexible and can be used for generating dataset for novel tasks
2) The authors demonstrate that training on synthetic data outperforms training on original data
3) Diversity is a key limitation in synthetic datasets, however it seems like lexical diversity is maintained

**Reasons To Reject:**

There were no solid reasons to reject, but some minor points that can be raised

1) The one main concern is that TarGen requires quite a bit of customization for each task, in terms of the instructions you provide for each step
2) The comparison with other techniques is lacking a lot, and while there may not be an apple-apple comparison (seed-less nature, requires task specific samples and so on) it would be better to compare against other techniques to bring out the performance of TarGen

For instance, there are some recent prompting based approaches that can be contrasted against TarGen such as https://arxiv.org/abs/2404.00415  https://aclanthology.org/2023.emnlp-main.647.pdf

---

> ### Author Rebuttal · Authors · 2024-05-30
>
> * **R1.1 - Generalizability of the framework**:The basic framework proposed in this work is task and model-agnostic. As shown in appendix B.4 and B.8, the instructions for (RTE) vs (AXg) are different. While RTE is a relatively straightforward task, AXg is a complex NLU task. Despite the differences in the prompts of these tasks, the underlying TarGEN framework delineating the steps remains common. This proposed framework can be adapted to any task, based on the problem being evaluated. Given the parameters of a task - the label schema, and the problem statement, this framework is easy to adapt for multiple tasks of varying complexities. We would like to highlight that the simple nature of our framework allows for greater flexibility and control, making it well-suited to domain-specific or particularly complex tasks. We are happy to generate any other tasks that the reviewer deems necessary for the verification of this approach.
> * **R1.2 - Comparison with other methods**: We compare our framework with other popular data generation approaches in the table below. We clarify ambiguous column headers:
>   - Seedless: Whether the approach requires labeled task samples.
>   - Focused diversity: Whether the approach actively injects diversity, incentivizing samples generated across various contexts.
>   - Noise mitigation strategy: Whether the approaches actively mitigate noise, either algorithmically or through the inclusion of a module like our Self-Correction as part of their pipeline
>
> |Method|Seedless|Training required as part of pipeline|Focused diversity|Ability to generate samples for a complex task|Noise mitigation strategy|
> |-|-|-|-|-|-|
> |[Self Instruct](https://arxiv.org/pdf/2212.10560)|No|No|No|No|Yes|
> |[CODA](https://arxiv.org/abs/2404.00415)|No| No|No|No| No|
> |[Synthetic Data Generation](https://aclanthology.org/2023.emnlp-main.647.pdf)|Yes|No|No|No|No|
> |[ZeroGEN](https://arxiv.org/abs/2202.07922)| Yes|Yes|No|No|No|
> |[SuperGEN](https://arxiv.org/pdf/2202.04538)|Yes|Yes|No|No|No|
> |[SunGEN](https://arxiv.org/abs/2205.12679)|Yes|Yes|No|No|Yes|
> |[ProGEN](https://aclanthology.org/2022.findings-emnlp.269/)|Yes|Yes|No|No|Yes|
> |Ours (TarGEN)|Yes|No|Yes|Yes|Yes|
>
> * **R1.3 - Effect of Instruction**: Determining the dynamics of "varying the instructions" and its corresponding impact on downstream tasks are too broad to quantify precisely. We respectfully request the reviewer to elaborate more on this point, following which we would be happy to respond.

---

> > ### Author Response · Authors · 2024-06-03
> > **Follow up to the Rebuttal**
> >
> > Dear Reviewer,
> >
> > We trust this message finds you well. We are writing to follow up on the rebuttal response we submitted in response to your feedback. We are eager to receive your insights and comments on our response. If there are any areas that you still find unclear or require further clarification, we would be more than happy to provide additional information to ensure a comprehensive understanding of our work.
> >
> > Your feedback is of utmost importance to us, and we greatly appreciate your time and consideration in evaluating our research.
> >
> > Thank you for your attention, and we look forward to hearing from you soon.

---

> > > ### Comment · Reviewer_BhhP · 2024-06-04
> > > **Response To Rebuttal**
> > >
> > > Dear author,
> > >
> > > Thanks for the detailed rebuttal.
> > >
> > > 1) Regarding the generalizability of the framework, like I mentioned before, the approach is flexible in the sense that it provides a rough guideline/steps in generating data for novel tasks. However, to actually apply them, a user has to experiment with prompt engineering at each step that works best for their particular task. In that sense, the generalizability is still limited.
> > > 2) Thanks for providing the table of comparison with existing approaches, however I was referring to a comparison of scores obtained with those methods. I do notice that it is presented in response to Reviewer QmfM. Adding these comparisons will certainly make your claims better.
> > > 3) Regarding the last point, on effect of instruction. I would like to explain myself better, as this is also related to point 1) here.
> > >
> > > Say for MultiRC, the following instruction is being used in the paper
> > >
> > > ```
> > > Given a paragraph, frame a question that requires information from multiple sentences of
> > > the paragraph to be answered correctly. Then, generate a set of options. The correct answer may
> > > be a combination of one, some, or all options. Also include options that do not answer the above
> > > question. Finally, output the combination of options that form the correct answer.
> > > ```
> > >
> > > But I decide to provide a similar instruction like the one below
> > >
> > > ```
> > > Read the text thoroughly to understand its content and themes. Create a question that requires integrating information from multiple sentences of the text. For each question, provide four to five option sentences that might seem relevant. it is not necessary that a combination of sentences answer the question. The options should test the reader’s ability to synthesize and evaluate information from the text, not just recall facts.
> > > ```
> > >
> > > Now two points that I had previously raised
> > > 1) Would it be fair to say that for each task and step, one has to do fair amount of prompt engineering to be able to elicit outputs for a step in Targen?
> > > 2) Given two such varied but similar instructions, how much does it affect my downstream performance?

---

> > > > ### Author Response · Authors · 2024-06-06
> > > > **Response to the comment**
> > > >
> > > > We thank the reviewer for their feedback and clarification of their concerns.
> > > >
> > > > To understand the effect of prompt variations on different downstream tasks, we conducted an experiment where we generated 2 additional variations of the Step 3 prompt for CommitmentBank dataset. Llama-2 7b was fine-tuned on these alternative synthetic datasets alongside our initially generated synthetic dataset and evaluated on the original test set.
> > > > We use the following prompts for this experiment:
> > > >
> > > >
> > > > Variation 1: [from paper]
> > > > ```
> > > > The input contains a premise and a hypothesis.
> > > > 1. If the hypothesis logically follows the premise and the hypothesis can be derived from the information in the premise, Output “entailment”
> > > > 2. If the hypothesis directly contradicts information in the premise, Output “contradiction”
> > > > 3. If the hypothesis is unrelated to the premise, or cannot be sufficiently proven from the information in the premise, Output “neutral”
> > > > ```
> > > >
> > > > Variation 2:
> > > > ```
> > > > The input consists of a premise and a hypothesis.
> > > > Output "neutral" if the hypothesis is either unrelated to the premise or cannot be conclusively derived from it.
> > > > Output "contradiction" if the hypothesis directly opposes the information in the premise.
> > > > Output "entailment" if the hypothesis logically follows from and can be deduced from the premise.
> > > > ```
> > > >
> > > > Variation 3:
> > > > ```
> > > > The input consists of a premise and a hypothesis.
> > > > If the hypothesis is directly opposed to the information in the premise, output “contradiction.”
> > > > If the hypothesis logically follows from the premise and can be inferred from its information, output “entailment.”
> > > > If the hypothesis is unrelated to the premise or lacks sufficient evidence from the premise, output “neutral.”
> > > > ```
> > > >
> > > > The results of the experiments are attached below:
> > > >
> > > > | Dataset                               | Mean CosineSimilarity | Accuracy - LLama-2 7B |
> > > > | ------------------------------------- | --------------------- | --------------------- |
> > > > | Synthetic with variation 1 [in paper] | 0.56                  | 94.85                 |
> > > > | Synthetic with variation 2            | 0.42                  | 93.27                 |
> > > > | Synthetic with variation 3            | 0.47                  | 94.14                 |
> > > >
> > > >
> > > > We see from the above results that there is slight variation in the final results and similar mean cosine similarity.
> > > >
> > > > We would also like to highlight that the motivation behind proposing a clearly-defined, task-agnostic guiding framework such as TarGEN is to constrain the prompt generation to a series of simpler, clearly defined subtasks. The cascaded stepwise approach has the following advantage: At each step, the subtask for which we prompt the model is straightforward with very little room for misinterpretation and errors. This reduces the prompt engineering effort significantly, especially compared to the human cost of engineering a single prompt that completely and accurately models a complex task. Any prompt that adheres to the instructions for that particular step in the framework is likely to generate similar outcomes, making this framework task- and prompt- agnostic.
> > > >
> > > > Additionally - the constrained, straightforward nature of the task prompt at each step makes it less likely that any variance in prompt would cause a significant change in outcome, given the narrower scope of the objective at each step and the fact that the outcomes are grounded in the response to the previous step. Furthermore, any slight deviances could be attributable to the randomness inherent in inferring with LLMs, but we would not expect to see a significant difference in quality or diversity of the output samples.
> > > >
> > > > If there are any areas that you still find unclear or require further clarification, we would be more than happy to provide additional information to ensure a comprehensive understanding of our work. Thank you for your attention, and we look forward to hearing from you soon.

---

> > > > > ### Comment · Reviewer_BhhP · 2024-06-06
> > > > > **Response To Rebuttal**
> > > > >
> > > > > Dear authors,
> > > > >
> > > > > Thanks for conducting these additional experiments regarding the variability of prompts. I appreciate these efforts and believe adding them will result in a more meaningful analysis.
> > > > >
> > > > > Given the fact that you have conducted stronger baseline comparison and ablation on prompt variability, I am inclined to increase the score.

---

### Decision · Program_Chairs · 2024-07-10

**Decision:**

Accept

**Comment:**

The authors presented TarGen, a novel framework for generating high-quality synthetic datasets using LLMs without the need for specific task instances, enhancing its applicability to novel or highly domain-specific tasks.

Strengths:
1. Impact: TarGen's seedless nature makes it applicable for various tasks, including novel and domain-specific ones without existing data instances (Reviewer BhhP, QmfM, b77X)
2. Quality: TarGen's use of self-correction module ensures reliable labels, and promotes diversity and complexity in the generated data (Reviewer BhhP, oq9P).
3. Evaluation: Reviewers appreciate the rigorous evaluation across tasks, including human validations (Reviewer QmfM, b77X).

Weakness -- Reviewers mostly questioned two things:
1. Whether the reliance on sophisticated prompting limit the generalizability of the approach (oq9P, BhhP)
2. Comparative Analysis: The paper lacks extensive comparative analysis with other existing methods and detailed ablation studies to elucidate the importance of each TarGEN component (All reviewers).

In their rebuttal, the authors added extensive experiments to address both questions, and as a result reviewers all feel convinced and the final rating are all above the acceptance threshold. As such, I recommend acceptance of the paper.